# ENHANCING GRAPH NEURAL NETWORKS WITH QUANTUM COMPUTED ENCODINGS

## ABSTRACT

Transformers are increasingly employed for graph data, demonstrating competitive performance in diverse tasks. To incorporate graph information into these models, it is essential to enhance node and edge features with positional encodings. In this work, we propose novel families of positional encodings tailored for graph transformers. These encodings leverage the long-range correlations inherent in quantum systems, which arise from mapping the topology of a graph onto interactions between qubits in a quantum computer. Our inspiration stems from the recent advancements in quantum processing units, which offer computational capabilities beyond the reach of classical hardware. We prove that some of these quantum features are theoretically more expressive for certain graphs than the commonly used relative random walk probabilities. Empirically, we show that the performance of state-of-the-art models can be improved on standard benchmarks and large-scale datasets by computing tractable versions of quantum features. Our findings highlight the potential of leveraging quantum computing capabilities to potentially enhance the performance of transformers in handling graph data.

## 1 INTRODUCTION

Graph machine learning (GML) is an expanding field of research with applications in chemistry (Gilmer et al., 2017), biology (Zitnik et al., 2018), drug design (Konaklieva, 2014), social networks (Scott, 2011), computer vision (Harchaoui & Bach, 2007) and science (Sanchez-Gonzalez et al., 2020; Xu et al., 2018). In the past few years, significant effort has been put into the design of Graph Neural Networks (GNNs) (Hamilton). The objective is to learn suitable representations that enable efficient solutions to the original problem.

To that end, a large number of models have been developed in the past few years (Kipf & Welling, 2016; Hamilton et al., 2018; Veličković et al., 2018). While the prevalent approach for constructing GNNs relies on the Message Passing (MP) mechanism (Gilmer et al., 2017), this approach exhibits several recognized limitations, with the most significant being its theoretical expressivity. Indeed, two graphs that are indistinguishable via the Weisfeiler-Lehman (WL) test will lead to the same MP Neural Network (MPNN) output (Morris et al., 2019). Another limitation arises from the fact that MPNNs are more effective when dealing with homophilic data. This is based on the underlying assumption that nodes that are similar, either in structure or features, are more likely to be related. A study by (Zhu et al., 2020) demonstrates that MPNNs encounter difficulties when applied to heterophilic graphs. Finally, MPNNs are prone to over-smoothing (Chen et al., 2020) as well as over-squashing (Topping et al., 2021). These latter aspects constitute serious limitations for the datasets that exhibit long-range dependencies (Dwivedi et al., 2022b).

The research community is actively exploring solutions to address these limitations. The key idea is to expand aggregation beyond neighbouring nodes by incorporating information related to the entire graph or a more extensive portion of it. Graph Transformers were created according to these requirements, with success on standard benchmarks (Ying et al., 2021; Rampášek et al., 2022). Among the myriad of proposed architectures, the Graph Inductive Bias Transformer (GRIT) (Ma et al., 2023) stands out for its impressive generalization capacity. This stems from its independence from the MP mechanism and its utilization of multiple positional encodings in its architecture. While these features make it a good candidate for overcoming the aforementioned limitations, the authors relied on discrete $k$-step random walks to initialize the PE tensor. These random walks constitute

to this day the most widely adopted choice (He et al., 2023; Dwivedi et al., 2022a), since the main alternative, the matrix of the Laplacian eigenvectors, is invariant by sign flip of each vector, resulting in $2^k$ possible choices.

The goal of this work is to leverage new types of structural features emerging from quantum physics as positional encodings. The rapid development of quantum computers during the previous years provides the opportunity to compute features that would be otherwise intractable. These features contain complex topological characteristics of the graph, and their inclusion has the potential to enhance the model's quality, reduce training or inference time, and decrease energy consumption.

The paper is organized as follows: In Section 2, we provide a concise overview of the existing research on graph transformers, along with references to the latest developments in quantum graph machine learning. Section 3 delves into the core theoretical aspects of this work. It covers quantum mechanics basics for readers unfamiliar with the topic, details the way to construct a quantum state from a graph and explains why quantum states can provide relevant information that is hard to compute with a classical computer. Additionally, we introduce our central proposal, a framework for both static and dynamic positional encoding based on quantum correlations. Finally, Section 4 presents the outcomes of our numerical experiments and includes discussions of the results.

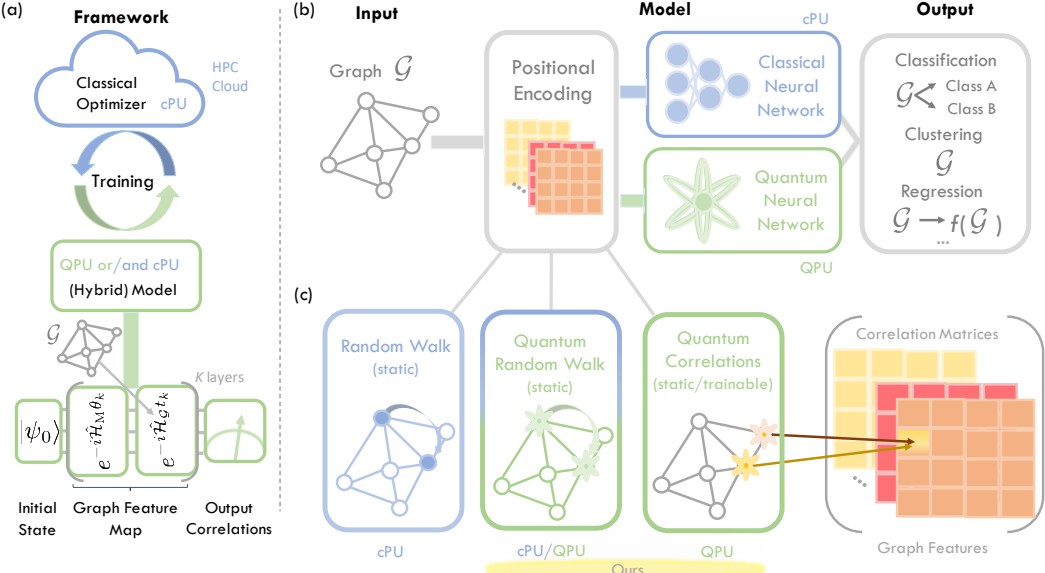

Figure 1: Summary of our method. **(a)** Our hybrid quantum-classical framework utilizes a classical computer for parameter optimization and employs a hybrid model using a Quantum Processing Unit (QPU) and a CPU and/or GPU, denoted as classical Processing Unit (cPU). In our quantum graph NN, we initialize QPU at a quantum state $|\psi_0\rangle$, apply a mixing Hamiltonian $\hat{\mathcal{H}}_M$ evolution for a duration $\theta$, and utilize a Hamiltonian $\hat{\mathcal{H}}_\mathcal{G}$ evolution for the graph feature map with a duration $t$. $K$ layers are used to obtain a sufficiently expressive quantum model. Finally, the output is obtained by measuring correlators, e.g., $\langle Z_i Z_j \rangle$. See Section 3.1 for details. **(b)** Static or trainable PE is constructed for a graph $\mathcal{G}$ via **(c)** (quantum) random walk (static PE) or a quantum graph NN (static/trainable PE), which computes quantum correlations. Note that our PEs are not restricted to classical models (such as the transformer studied in this work) but are also applicable to all quantum models.

## 2 RELATED WORKS

### 2.1 GRAPH TRANSFORMERS

Efforts have been made in the community to go beyond MPNNs due to several issues (Zhu et al., 2020; Chen et al., 2020; Topping et al., 2021). Inspired by the success of transformers in natural language processing (Vaswani et al., 2017; Alayrac et al., 2022), new architectures of GNNs have

been proposed to allow an all-to-all aggregation between the nodes of the graphs, and called graph transformers (GT) (Dwivedi & Bresson, 2020; Dwivedi et al., 2021; Rampášek et al., 2022; Kreuzer et al., 2021; Zhang et al., 2023; Ma et al., 2023). However, due to the quadratic cost of computing the attention process, they are not applicable to large-scale graphs of millions of nodes and more. It has been shown that GTs that include graph inductive biases such as MP modules perform better than those that do not (Rampášek et al., 2022; Ma et al., 2023).

## 2.2 Positional and Structural Encoding

Positional or structural embeddings are features computed from the graph that are concatenated to original node or edge features to enrich GNN architectures (either MPNN or GT). These two terms are used interchangeably in the literature and we denote them as "positional encodings" (PEs) in the rest of this work. PEs can include random walk probabilities (Rampášek et al., 2022; Ma et al., 2023), spectral information (Dwivedi et al., 2020; Rampášek et al., 2022; Kreuzer et al., 2021), shortest path distances (Li et al., 2018), or heat kernels (Mialon et al., 2021). They can also be learned (Dwivedi et al., 2021). We detail below the most common PEs used in the literature.

**Laplacian Eigenvectors.** The spectral information of the graph can be used as PE, more precisely the eigenvectors of the Laplacian matrix. If one takes a line graph, it almost corresponds to positional embeddings in the transformer architecture for sequences. The main issue of this encoding is to ensure that the model remains invariant by changing the sign of eigenvectors, which has been solved by (Lim et al., 2022).

**Relative Random Walk Probabilities (RRWP).** The authors of (Ma et al., 2023) introduced the RRWP with which they initialize their model. For a graph $\mathcal{G}$, let $A$ be the adjacency matrix and $D$ the degree matrix. Let $P$ be a 3 dimensional tensor such that $P_{k,i,j} = (M^k)_{ij}$ with $M = D^{-1}A$. For each pair of node $(i, j)$, we associate the vector $P_{:,i,j}$, i.e., the concatenation of the probabilities for all $k$ to get from node $i$ to node $j$ in $k$ steps in a random walk. $P_{:,i,i}$ is the same as the Random Walk Structural Encodings (RWSE) defined in (Rampášek et al., 2022). The authors of (Ma et al., 2023) highlight the benefits of RRWP. They prove that the Generalized Distance WL (GD-WL) test introduced by (Zhang et al., 2023) with RRWP is strictly more powerful than GD-WL test with the shortest path distance, and they prove universal approximation results of multi-layer perceptrons (MLP) initialized with RRWP.

## 2.3 Quantum Computing for Graph Machine Learning

Using quantum computing for Machine Learning on graphs has already been proposed in several works, as reviewed in (Tang et al., 2022). The authors of (Verdon et al., 2019) realized learning tasks by using a parameterized quantum circuit depending on a Hamiltonian whose interactions share the topology of an input graph. Comparable ideas were used to build graph kernels from the output of quantum procedures, for photonic (Schuld et al., 2020) as well as neutral atom quantum processors (Henry et al., 2021). The latter was successfully implemented on quantum hardware (Albrecht et al., 2023). The architectures proposed in these papers were entirely quantum and only relied on classical computing for the optimization of variational parameters. By contrast, in what we propose here quantum dynamics only plays a role in the aggregation phase of a larger entirely classical architecture. Such a hybrid model presents the advantage of gaining access to hard-to-access graph topological features through quantum dynamics while benefiting from the power of well-known existing classical architectures.

# 3 Methods and theory

In this section, we outline the process of mapping graphs to a quantum state of a QPU. To extract graph features, we introduce correlators and define the concept of the ground state for a quantum graph representation. Finally, we explore an alternative approach for extracting graph features using quantum random walks (QRW) and their advantages over classical analogues.

### 3.1 Quantum Graph Machine Learning

**The graph as a quantum state.** We explain in this subsection how to create a quantum state that contain relevant information about the graph. More details about quantum information processing can be found in (Nielsen & Chuang, 2002). The *quantum state* $|\psi\rangle$ of a system of $N$ qubits can be represented as a vector of unit norm in $\mathbb{C}^{2^N}$. Quantum states are modified through the action of *operators*, that can be represented as hermitian matrices of size $2^N \times 2^N$. Its dynamics obeys the Schrödinger equation $-i\frac{d|\psi\rangle}{dt} = \hat{\mathcal{H}}|\psi\rangle$, where the operator $\hat{\mathcal{H}}$ is the *Hamiltonian* of the system, with solution $|\psi(t)\rangle = \mathcal{T}\exp\left[-i\int_0^t \hat{\mathcal{H}}(\tau)d\tau\right]|\psi(0)\rangle$, with $\mathcal{T}$ the time-ordering operator. An operator $\hat{\mathcal{O}}$ that can be measured is called an *observable*, and its eigenvalues correspond to possible outcome of its measurement. Its expectation value on the quantum state $|\psi\rangle$ is the scalar $\langle\hat{\mathcal{O}}\rangle = \langle\psi|\hat{\mathcal{O}}|\psi\rangle$, where $\langle\psi|$ is the conjugate transpose of $|\psi\rangle$. The *Pauli matrices* are defined as follows: $I = \left(\begin{smallmatrix} 1 & 0 \\ 0 & 1 \end{smallmatrix}\right), X = \left(\begin{smallmatrix} 0 & 1 \\ 1 & 0 \end{smallmatrix}\right), Y = \left(\begin{smallmatrix} 0 & -i \\ i & 0 \end{smallmatrix}\right), Z = \left(\begin{smallmatrix} 1 & 0 \\ 0 & -1 \end{smallmatrix}\right)$.

They form a basis of hermitian matrices of size $2 \times 2$. A *Pauli string* of size $N$ is an operator that can be written as the Kronecker product of $N$ Pauli matrices. We will note Pauli strings by their non-trivial Pauli operations. For instance, in a system of 5 qubits, $X_0 Y_3 = X \otimes I \otimes I \otimes Y \otimes I$. We associate a graph $\mathcal{G}(\mathcal{V}, \mathcal{E})$, to a quantum state $|\psi_\mathcal{G}\rangle$ of $|\mathcal{V}|$ qubits containing information about $\mathcal{G}$ via a hamiltonian $\hat{\mathcal{H}}_\mathcal{G}$ of the form

$$\hat{\mathcal{H}}_\mathcal{G} = \sum_{(i,j)\in\mathcal{E}} \hat{\mathcal{H}}_{ij} \tag{1}$$

where $\hat{\mathcal{H}}_{ij}$ is an Pauli string acting non-trivially on $i$ and $j$ only. We will be focusing on the Ising hamiltonian $\hat{\mathcal{H}}^I = \sum_{(i,j)\in\mathcal{E}} Z_i Z_j$ and the XY hamiltonian $\hat{\mathcal{H}}^{XY} = \sum_{(i,j)\in\mathcal{E}} X_i X_j + Y_i Y_j$. We will note $|0\rangle$ and $|1\rangle$ the two eigenstates (or eigenvectors) of $Z$ with respective eigenvalues 1 and -1, and we will use $\left\{|\mathbf{b}\rangle = \bigotimes_{i=1}^N |b_i\rangle\right\}_{\mathbf{b}\in\{0,1\}^N}$ as a basis of the $2^N$-dimensional space of quantum states.

We consider the quantum state obtained by alternated action of $p$ layers of $\hat{\mathcal{H}}_\mathcal{G}$ and a *mixing* hamiltonian $\hat{\mathcal{H}}_M$ (that doesn't commute with $\hat{\mathcal{H}}_\mathcal{G}$, for instance $\hat{\mathcal{H}}_M \propto \sum_i Y_i$)

$$|\psi_\mathcal{G}(\boldsymbol{\theta})\rangle = \prod_{k=1}^p \left(e^{-i\hat{\mathcal{H}}_M\theta_k}e^{-i\hat{\mathcal{H}}_\mathcal{G}t_k}\right) e^{-i\hat{\mathcal{H}}_M\theta_0}|\psi_0\rangle, \tag{2}$$

where $\boldsymbol{\theta} = (\theta_0, t_0, \theta_1, t_1, \dots \theta_p)$ is a real vector of parameters. The choice of these states is motivated by their similarity with the *Trotterized* dynamics of a lot of quantum systems(Suzuki, 1976).

**Correlation.** The correlations (or *correlators*) $C_{ij}$ of local operators $\hat{\mathcal{O}}_i$ and $\hat{\mathcal{O}}_j$ acting respectively on qubits $i$ and $j$ can be defined either as the expectation value of their product $\langle\hat{\mathcal{O}}_i\hat{\mathcal{O}}_j\rangle$, or their covariance $\langle\hat{\mathcal{O}}_i\hat{\mathcal{O}}_j\rangle - \langle\hat{\mathcal{O}}_i\rangle\langle\hat{\mathcal{O}}_j\rangle$ (note that the orders matters if $\hat{\mathcal{O}}_i$ and $\hat{\mathcal{O}}_j$ don't commute). In the rest of the paper, we will indifferently call correlation the two former expressions, and give precisions when necessary. We will be focusing on the case where $\hat{\mathcal{O}}_i$ is a Pauli string of length 1 (*i.e.* $X_i$, $Y_i$ or $Z_i$).

**Ground state.** The ground state of a system is defined as the lowest-energy eigenstate of its hamiltonian (when it is degenerate, one considers the *ground state manifold* $\mathbb{H}_{GS}$). Ground state properties are widely studied in many-body physics and their properties depend on the topology of the graph. Preparing this state is the purpose of quantum annealing (Das & Chakrabarti, 2008). When using neutral atom quantum processors (Henriet et al., 2020), one can natively address hamiltonians of the form $\hat{\mathcal{H}}_\mathcal{G} = \sum_{(i,j)\in\mathcal{E}} J_{ij}(Z_i - \alpha_i I)(Z_j - \alpha_j I)$, with $\alpha_i$ real coefficients. Its eigenstates are the basis states $|\mathbf{b}\rangle$ described above. In the case where $\alpha_i = 1 - \delta/(2z_i)$ with $z_i = \sum_{j|(i,j)\in\mathcal{E}} J_{ij}$ and $J_{ij} = 1/4$, the eigenenergies (or eigenvalues) are $E(\mathbf{b}) = \sum_{i,j\in\mathcal{E}} b_i b_j - \delta\sum_{i=1}^N b_i$. When $0 < \delta < 1$, this is the cost function associated with the maximum independent set problem, a NP-hard problem (Garey & Johnson, 1979). In the absence of degeneracy-lifting or symmetry-breaking effects, a quantum annealing scheme would prepare a symmetric, equal-weight superposition of all maximum independent sets. With that in mind, we will call *ground state of the graph* the state $|\psi_{GS}\rangle = \frac{1}{\sqrt{|\mathbb{H}_{GS}|}}\sum_{\mathbf{b}\in\mathbb{H}_{GS}} |\mathbf{b}\rangle$.

**Classical and Quantum Walks.** Quantum walks, as introduced by (Aharonov et al., 1993), differ fundamentally from classical random walks by evolving through unitary processes, allowing for interference between different trajectories. These walks manifest in two primary types: *continuous-time quantum walks* (CQRW) (Farhi & Gutmann, 1998; Rossi et al., 2017) and *discrete-time quantum walks* (DQRW) (Lovett et al., 2010). Discrete classical random walks on $\mathcal{G}(\mathcal{V}, \mathcal{E})$ use the probability matrix $M = D^{-1}A$ for node transitions over a walk of length $K$, resulting in the probability distribution $P_K = M^K P_0$ (Aharonov et al., 2001). Note that this approach is utilized in RRWP encodings (Ma et al., 2023). In the continuous case, CQRW can be viewed as a natural extension of continuous-time classical random walks (CRW). In CRW, the probability of a walker being at vertex $i$ and time $t$ is represented as $p_i(t)$, which follows the differential equation $\frac{\mathrm{d}}{\mathrm{d}t} p_i(t) = -\sum_j G_{ij} p_j(t)$. Here, the infinitesimal generator $G_{ij} = -\gamma$ if an edge exists between nodes $i$ and $j$, and 0 otherwise, with diagonal elements $G_{ii} = k_i \gamma$ determined by the node degree $k_i$. Considering now a quantum evolution with a graph Hamiltonian $\hat{\mathcal{H}}_\mathcal{G}$, given a $2^N$-dimensional Hilbert space of $N$ qubits, the Schrödinger equation which governs the evolution of a quantum state $|\psi_\mathcal{G}\rangle$ when projected onto a state $|i\rangle$ is given as

$$\underbrace{i \frac{\mathrm{d}}{\mathrm{d}t} \langle i | \psi_\mathcal{G}(t) \rangle = \sum_j \langle i | \hat{\mathcal{H}}_\mathcal{G} | j \rangle \langle j | \psi_\mathcal{G}(t) \rangle}_{\text{Quantum}} \longleftrightarrow \underbrace{\frac{\mathrm{d}}{\mathrm{d}t} p_i(t) = -\sum_j G_{ij} p_j(t)}_{\text{Classical}}. \tag{3}$$

Note the similarity between the differential equations of CQRW and CRW. A quantum analogue of CRW can be obtained by taking $\langle i | \hat{\mathcal{H}}_\mathcal{G} | j \rangle = G_{ij}$. The probabilities are preserved as the sum of amplitude squared, $\sum_i |\langle i | \psi_\mathcal{G}(t) \rangle|^2 = 1$, in the quantum case, instead of $\sum_i p_i(t) = 1$ in the classical case. This difference between evolution of probabilities(which are real) and evolution of amplitudes(which are complex) leads to interesting differences between dynamics of classical and quantum walks. Using this formalism, any quantum evolution can be thought of as a CQRW (Childs et al., 2002). Notably, quantum walks have demonstrated exponential hitting time advantage for graphs like hypercubes (Kempe, 2002) and glued binary trees (Childs et al., 2003). These results have been recently extended for more general hierarchical graphs (Balasubramanian et al., 2023). For an overview, refer to (Kempe, 2003).

### 3.2 POSITIONAL ENCODINGS WITH QUANTUM FEATURES

In this section, we detail our proposals to incorporate quantum features in GNN models, and we discuss the potential benefits and drawbacks. Our methods can be roughly divided in two categories: quantum features that are used as static positional encodings and are precomputed at the begining of the procedure and quantum features that can be dynamically trained.

#### 3.2.1 STATIC POSITION ENCODING

**Eigenvectors of the correlation on the ground state.** We propose to use the correlation matrix $C_{ij} = \langle Z_i Z_j \rangle$ on the ground state of the graph defined in Sec. 3.1. Since this matrix is symmetric with non-negative eigenvalues, it can formally be used in the same place as the Laplacian matrix in graph learning models. Hence, we use the eigenvectors of this correlation matrix in the same way Laplacian eigenvectors (LE) are used in other architectures of graph transformers. Instead of taking the eigenvectors with the lowest eigenvalues as for the Laplacian eigenmaps, we take the ones with highest eigenvalues, since they are the ones in which most of the information about the correlation matrix is contained. We expect to face the same challenges about the sign ambiguity (Dwivedi et al., 2021; Kreuzer et al., 2021), and to implement the same techniques to alleviate them (Lim et al., 2022).

$k$-**particles quantum random walks ($k$-QRW).** In this work, we introduce the $k$-particles (or walkers) random walk positional encoding, that can be can be obtained using $\hat{\mathcal{H}}^{XY}$. We note $\hat{\mathcal{H}}_k^{XY}$ the XY hamiltonian restricted to the $k$ particles subspace $\mathbb{H}_k$ (*i.e.* the span of states $|\mathbf{b}\rangle$ of hamming weight $k$, noted $|i_1 \dots i_k\rangle$, parameterized by $k$ integers $i_1 \dots i_k \in \{0, 1\}^k$). For a 1-particle QRW, we calculate the probability $[X^{(1)}(t)]_{ij} = |\langle j| e^{-i\hat{\mathcal{H}}_1^{XY} t} |i\rangle|^2$ to find particle at node $j$ coming from node $i$ after time $t$. Similarly for a 2-particle QRW, we calculate $[X^{(2)}(t)]_{ij} = |\langle ij| e^{-i\hat{\mathcal{H}}_2^{XY} t} |\psi_i\rangle|^2$, where $|i, j\rangle \in \hat{\mathcal{H}}_2^{XY}$ is the state with walkers at nodes $i$ and $j$ and $|\psi_i\rangle \in \hat{\mathcal{H}}_2^{XY}$ the initial state. As

choices for the initial distribution we propose to use some localised state $|\psi_{\text{init}}\rangle \propto |ij\rangle$, or the uniform distribution over all pairs of nodes $|\psi_{\text{init}}\rangle \propto \sum_{(i,j)\in\mathcal{V}^2|i\neq j} |ij\rangle$, or the uniform distribution over the edges of the original graph $|\psi_{\text{init}}\rangle \propto \sum_{(i,j)\in\mathcal{E}} |ij\rangle$. From these we obtain the positional encodings using $\mathbf{P}_{ij} = [I, X^{n_w}(t_1), X^{n_w}(t_2)...X^{n_w}(t_K)]_{ij}$, where $n_w = 1, 2$ is the number of walkers. From the symmetries of $\hat{\mathcal{H}}^{XY}$, this 2-QRW can be viewed as a 1-QRW on the set of 2-particle states (see A.3.2). From there, we consider a *discrete* 2-particle quantum-inspired RW (2-QiRW) encoding that reads $\mathbf{P}_{ij} = \left[\langle ij|(\hat{\mathcal{H}}_2^{XY})^k|\psi_{\text{init}}\rangle|k\in[0,K]\right]_{ij}$.

### 3.2.2 LEARNABLE POSITIONAL ENCODINGS

Here we consider a specific case of equation 2, with $p = 1$, $\theta_0 = -\theta_1 = \theta$, $\hat{\mathcal{H}}_M \propto \sum_{i\in\mathcal{V}} Y_i$ and $\hat{\mathcal{H}}_\mathcal{G} = \sum_{(i,j)\in\mathcal{E}} Z_i Z_j - \delta \sum_{i\in\mathcal{V}} Z_i$. A similar setting was implemented on neutral atom QPU in (Albrecht et al., 2023), where $\hat{\mathcal{H}}_M \propto \sum_{i\in\mathcal{V}} X_i + \varepsilon\hat{\mathcal{H}}_\mathcal{G}$ (with $\varepsilon \lesssim 1$) due to hardware constraints. We then use the covariance matrix of the number of occupation observable $\hat{\mathcal{O}}_i = \frac{1}{2}(I - Z_i)$, which equals one when the $i-$th atom is in its excited state, and 0 otherwise. This is one of the few particular cases where we can recover a closed formula for the correlation matrix used as PE, see Appendix A.2.1 for its full expression.

The goal in this section is to learn the positional encoding by training a GNN, or any permutation equivariant NN, to find an optimised value of the parameters ($\theta$, $t$ and $\delta$) involved in our PE. The training of this module is carried out jointly with that of the transformer layers, and the input PE is updated after each backward pass. This allows a custom value of the parameters for each graph in the dataset. To this end, we train a GNN $P_{\mathbf{W}}(\mathbf{A}, \mathbf{X})$, with $\mathbf{A}$ being the adjacency matrix, $\mathbf{X}$ the feature matrix of the nodes in the graph, and $\mathbf{W}$ the parameter set of the NN. We obtain the PE parameters as $(\theta||t||\delta) = P_{\mathbf{W}}(\mathbf{A}, \mathbf{X}) \in \mathbb{R}^{3\times k}$ (we learn $k$ triples, encoding as many correlations matrices, which we concatenate in one tensor as it is done in the original GRIT paper). Here we chose $\mathbf{X}$ as the pairwise graph distance matrix between the nodes, such that we only consider its structural features. Since the positional encoding discussed here is obtained from a special case of QAOA, a key aspect of this approach concerns the initial values taken by $\theta, t$ and $\delta$ (Egger et al., 2021). Among the many possible initialization protocols, the authors in (Jain et al., 2022) used GNN to find a warm start in the non-convex energy landscape, prior to an optimization through quantum annealing, for the Max-Cut problem. In our case, the training is carried out classically (only because we recover a closed formula), and its extension as a quantum NN module in a transformer-like architecture is yet to be investigated in future works.

### 3.3 GNN MODELS WITH QUANTUM AGGREGATION

In the previous subsection, we explained how to integrate positional encodings coming from a quantum processing unit into a graph transformer of graph neural network model. In this subsection, we propose to directly include the quantum correlations as a trainable part of the model. Given $H^l$ the node features matrix at the layer $l$, the node features matrix at the next layer is computed with the following formula :

$$H^{l+1} = \sigma((A(\theta)H^l||H^l)W) \tag{4}$$

where $A(\theta)$ is the quantum attention matrix. Given a quantum graph state, we compute for every pair of nodes $(i, j)$ the vector of 2-bodies observables $C_{ij} = [\langle Z_i Z_j\rangle, \langle X_i X_j\rangle, \langle Y_i Y_j\rangle, \langle X_i Z_j\rangle, \langle X_i Y_j\rangle, \langle Y_i Z_j\rangle, \langle X_j Z_i\rangle, \langle X_j Y_i\rangle, \langle Y_j Z_i\rangle]^T$. The quantum attention matrix is computed by taking a linear combination of the previous correlation vector and optionally a softmax over the lines. More details are given in the Appendix B.

### 3.4 THEORETICAL ARGUMENTS FOR QRWS

**Two-interacting-particle QRW are more expressive than RRWP.** WL tests (Weisfeiler & Leman, 1968) and their extensions (Morris et al., 2019; Grohe & Otto, 2015; Grohe, 2017) are widely used algorithms for distinguishing graphs (Graph Isomorphism). A standard WL test generates node

colorings as a function of the node neighbourhood. An extension of this called the generalized distance WL (GD-WL) test had been introduced by (Zhang et al., 2023) as a generalization of the WL isomorphism test. Let $\mathcal{G}$ be a graph on a set of vertices $\mathcal{V}$ and let $d(u, v)$ be the distance between nodes $u, v$. Just like the WL test, the GD-WL test starts with an initial color $h_0(u)$ for a node u. Then at iteration $l$, the color $h_l(u)$ is computed by $h_l(u) = \text{HASH}(\{(h_{l-1}(v), d(u, v)), v \in \mathcal{V}\}$ where HASH is a hash function. The algorithm stops when the colors don't change after an iteration. The authors of Zhang et al., 2023 show that if equipped with some distances such as the shortest path distance (SPD), the GD-WL test is strictly more powerful than the WL test. The authors of (Ma et al., 2023) show that the GD-WL test equipped with RRWP is strictly more powerful than the GD-WL test with the SPD distance. We can show however (see Appendix A.3.1 for a proof), that GD-WL test with RRWP embeddings cannot distinguish non-isomorphic strongly regular graphs (see (Gamble et al., 2010) for the definition of strongly regular graphs) :

**Theorem 1** *GD-WL test with RRWP embedding fails to distinguish non isomorphic strongly regular graphs. 2 particle-QRW can distinguish some of them.*

## 4 EXPERIMENTS

We performed several experiments to assess the capabilities of quantum features to improve existing GNN models. We report in the following subsections the most significant results. Less conclusive experiments are detailed in Appendices C.4, C.3.

### 4.1 EXPERIMENTS ON RW MODELS

In this subsection, we test concatenating the QRW encodings to the RRWP in the GRIT model (Ma et al., 2023). We compute the (continuous) 1-CQRW for $K$ random times and the discrete 2-QRW for $K$ steps. Those encodings are computed numerically since they are still tractable for graphs below 200 nodes compared to the higher order $k$-QRW ones. We benchmark our method on 7 datasets from (Dwivedi et al., 2020), following the experimental setup of (Rampášek et al., 2022) and (Ma et al., 2023). Our method is compared to many other architectures and the results directly taken from (Ma et al., 2023). We do not perform an extensive hyperparameter search for each architecture and only run ourselves the GRIT model by taking the same hyperparameters as the authors. The experiments are done by building on the codebase of (Ma et al., 2023) which is itself built on (Rampášek et al., 2022). More details about the protocol and hyperparameters can be found in C.1, and more details about the datasets can be found in Appendix D.1. The results are included in Table 1. Our methods performs better on ZINC, MNIST and CIFAR10 than all others, and comes second for PATTERN and CLUSTER. We also benchmark our methods on large-scale datasets, ZINC-full (a bigger version of ZINC (Irwin et al., 2012)) and PCQM4MV2 (Hu et al., 2021). As before we only compute GRIT and report other results from (Ma et al., 2023). Our methods performs the best among all others.

### 4.2 EXPERIMENTS ON LEARNING THE POSITIONAL ENCODING

Here we discuss results related to the learned parameters of the quantum PE, which were mainly run on the ZINC dataset. We observe from the comparison of these results, displayed in the right part of table 2, and those of random parameters (table 1) that the performances are significantly lower in this case. The main reason for this is the incapacity of the proposed model, to efficiently explore the space of the encoded parameters $\theta$, $t$ and $\delta$. More details about this are provided in the Appendix A.1 Given such limitations, the question arises as to whether the proposed positional encoding, via learned or randomly fixed parameters, plays any role in the training of the transformer model, or whether the attention scheme proposed in (Ma et al., 2023) along with the graph external features is enough to obtain the same performances. For that we compare our results with three different GRIT+RRWP models on randomized Zinc datasets. In the first one, we remove the structural information while keeping the degree sequence in each graph (configuration model of the graph (Newman, 2010)), in the second we remove any trace of structural information by randomly replacing each graph in the dataset by a random graph with the same number of nodes and edges, and in the final case we keep the structural information but randomly permute the feature vectors of each graph, such that we isolate the contribution of the structural features from that of the external features. In each case, we benchmark these results using a 2 layered graph neural network (with GAT convolutional layers), to

Table 1: Test performance in five benchmarks from (Dwivedi et al., 2020). We show the mean ± s.d. of 4 runs with different random seeds as in (Ma et al., 2023). Highlighted are the top first, second, and third results. Models are restricted to ∼ $500K$ parameters for ZINC, PATTERN, CLUSTER ∼ $100K$ for MNIST and CIFAR10. We compare our model to our run of GRIT and indicate the results obtained by the authors for information. Figures other than the last 3 lines are taken from (Ma et al., 2023). Models in **bold** are our models.

| Model | ZINC | MNIST | CIFAR10 | PATTERN | CLUSTER |
|---|---|---|---|---|---|
| | MAE↓ | Accuracy↑ | Accuracy↑ | Accuracy↑ | Accuracy↑ |
| GCN | $0.367 \pm 0.011$ | $90.705 \pm 0.218$ | $55.710 \pm 0.381$ | $71.892 \pm 0.334$ | $68.498 \pm 0.976$ |
| GIN | $0.526 \pm 0.051$ | $96.485 \pm 0.252$ | $55.255 \pm 1.527$ | $85.387 \pm 0.136$ | $64.716 \pm 1.553$ |
| GAT | $0.384 \pm 0.007$ | $95.535 \pm 0.205$ | $64.223 \pm 0.455$ | $78.271 \pm 0.186$ | $70.587 \pm 0.447$ |
| GatedGCN | $0.282 \pm 0.015$ | $97.340 \pm 0.143$ | $67.312 \pm 0.311$ | $85.568 \pm 0.088$ | $73.840 \pm 0.326$ |
| DGN | $0.168 \pm 0.003$ | — | $72.838 \pm 0.417$ | $86.680 \pm 0.034$ | — |
| GIN-AK+ | $0.080 \pm 0.001$ | — | $72.19 \pm 0.13$ | $86.850 \pm 0.057$ | — |
| SAN | $0.139 \pm 0.006$ | — | — | $86.581 \pm 0.037$ | $76.691 \pm 0.65$ |
| K-Subgraph SAT | $0.094 \pm 0.008$ | — | — | $86.848 \pm 0.037$ | $77.856 \pm 0.104$ |
| EGT | $0.108 \pm 0.009$ | $98.173 \pm 0.087$ | $68.702 \pm 0.409$ | $86.821 \pm 0.020$ | $79.232 \pm 0.348$ |
| GPS | $0.070 \pm 0.004$ | $98.051 \pm 0.126$ | $72.298 \pm 0.356$ | $86.685 \pm 0.059$ | $78.016 \pm 0.180$ |
| GRIT | $0.059 \pm 0.002$ | $98.108 \pm 0.111$ | $76.468 \pm 0.881$ | $87.196 \pm 0.076$ | $80.026 \pm 0.277$ |
| GRIT (our run) | $0.060 \pm 0.002$ | $98.164 \pm 0.054$ | $76.198 \pm 0.744$ | $90.405 \pm 0.232$ | $79.856 \pm 0.156$ |
| **GRIT 1-CQRW** | $0.058 \pm 0.002$ | $98.108 \pm 0.111$ | $76.347 \pm 0.704$ | $87.205 \pm 0.040$ | $78.895 \pm 0.1145$ |
| **GRIT 2-QiQRW** | $0.059 \pm 0.004$ | $98.204 \pm 0.048$ | $76.442 \pm 1.07$ | $90.165 \pm 0.446$ | $79.777 \pm 0.171$ |

Table 2: Left part : test performance on ZINC-full (Irwin et al., 2012) and PCQM4Mv2. For ZINC-full, we show the mean and s.d of 4 runs with different random seeds and we limit the model to ∼ $500K$ parameters. For PCQM4Mv2 we show the output of a single run due to computation time. We compare our model to our run of GRIT and indicate the results obtained by the authors for information. Figures other than GRIT are taken from (Ma et al., 2023). Highlighted are the top first, second, and third results. Right part: results related to the Isolation of the effects of structural and external graph features in the GRIT model, on the small zinc dataset. For the result with $^*$ only 55 epochs were used.

| Method | Model | ZINC-full (MAE ↓) | PCQM4Mv2 (MAE ↓) | Model | Zinc (MAE ↓) |
|---|---|---|---|---|---|
| | GIN | $0.088 \pm 0.002$ | $0.1195$ | — | — |
| | GraphSAGE | $0.126 \pm 0.003$ | — | — | — |
| MPNNs | GAT | $0.111 \pm 0.002$ | — | — | — |
| | GCN | $0.113 \pm 0.002$ | $0.1195$ | — | — |
| PE-GNN | SignNet | $0.024 \pm 0.003$ | — | — | — |
| | Graphormer | $0.052 \pm 0.005$ | $0.0864$ | **Rand. feats +RRWP** | $0.393 \pm 0.012$ |
| | Graphormer-URPE | $0.028 \pm 0.002$ | — | **Rand. struct. +RRWP** | $0.245 \pm 0.009$ |
| Graph | Graphormer-GD | $0.025 \pm 0.004$ | — | **Config. model+RRWP** | $0.156 \pm 0.004$ |
| Transformers | GPS-medium | — | $0.0858$ | **2 layers GAT+QCorr** | $0.134 \pm 0.017$ |
| | GRIT (Ma et al., 2023) | $0.023 \pm 0.001$ | $0.0859$ | **2 layers SAGE+QCorr** | $0.130 \pm 0.003$ |
| | GRIT (our run) | $0.025 \pm 0.002$ | $0.0842$ | **2 layers Transf. +QCorr** | $0.127 \pm 0.014$ |
| | **GRIT 1-CQRW (ours)** | $0.025 \pm 0.003$ | $0.0947^*$ | **2 layers GCN +QCorr** | $0.111 \pm 0.003$ |
| | **GRIT 2-QiQRW (ours)** | $0.023 \pm 0.002$ | $0.0838$ | — | — |

encode the parameters of the PE tensor. The results of these comparisons are showcased in the right part of table 2, and further details about the randomization process in Appendix A.2

## 4.3 SYNTHETIC EXPERIMENTS

In this section, we provide two examples of datasets with a binary graph classification task for which the use of the correlation matrix on the ground state as defined in 3.2.1 is more powerful than other commonly used features like the eigenvectors of the laplacian matrix or the diagonal of the random walk matrix. We name our datasets special pattern (S-PATTERN) and cross ladder (C-LADDER), and we provide a detail of their construction in appendices D.1.1 and D.1.2. The idea is to construct graphs that will exhibit very different Ising ground states but similar spectral properties or random walk transition probabilities. We illustrate the differences between the encodings in Appendix C.2. We train classical models on these datasets with LEs and random walk embeddings (RWSE) as node

features, and we compare it to the same models with eigenvectors of the correlation on the ground state. We experiment with GCN and GPS with many hyperparameters combinations. More details on the protocols can be found in C.2 We also benchmark the GRIT model with RRWP. The results are shown in table 3. The quantum encoding models achieve 100% accuracy in both cases whereas the models with LE or RWSE achieve only 57% maximum. The GRIT model achieves 78% for the S-PATTERN, and 100% for the C-LADDER.

Table 3: Results on synthetic data. We show the accuracy on the test set for each datasets. For each positional encoding, we show the score of the best model among all the combinations of hyperparameters tested. LE : Laplacian Eigenvectors, RWSE: Random Walk Structural Encodings, Q: Quantum, eigenvectors of the correlation on the ground state.

| Dataset | (GCN/GPS)-LE | (GCN/GPS)-RWSE | GRIT-RRWP | (GCN/GPS)-Q |
|---------|--------------|----------------|-----------|-------------|
| C-LADDER | 57 | 56 | 78 | 100 |
| S-PATTERN | 56 | 55 | 100 | 100 |

## 4.4 Discussion

We performed several experiments comparing the quantum encodings to the classical ones. Including the quantum walk features into state of the art models improves their performances on most of the datasets tested. It is not surprising that the method works well for datasets for which random walks are known to be relevant features like ZINC (Rampášek et al., 2022). We only limited ourselves to versions of quantum features that are efficiently computable, and we were able to show a small gain in performances compared to state of the art models. It is then believable that using quantum features that cannot be classically accessible could lead to a great improvement of models, if quantum hardware can be made widely available. We were able to engineer artificial datasets with for which classical approaches have difficulties to perform the associated binary classification tasks. GPS model fails both of the tasks whereas GRIT model is successful on the S-PATTERN task and mildly successful on the C-LADDER task. We think GRIT is successful on the S-PATTERN dataset because the degree distribution is different for the two classes, and GRIT specifically uses a degree scaler in its architecture. We have addressed, albeit at a preliminary stage, the issue of learning parameters related to the positional encoding. We have shown that even with non-optimal parameters (noted QCorr on the right-hand side of the table 2), the model still performs better than when undergoing structural randomization, which underlines the discriminative capacity of the QCorr positional encoder, and that the performance of the GRIT-QCorr model is not exclusively due to the graph external features.

## 5 Conclusion

In this paper we have investigated how quantum computing architectures can be used to construct new families of graph neural networks. This study involved measuring observables like correlations and probabilties for a quantum system whose hamiltonian has the same topology as the graph of interest. We then integrated these observables as positional encodings and used them in different classical graph neural network architectures. We also used them as attention layers in graph transformers. We proved that some positional encodings that use quantum features are theoretically more expressive than ones based on simple random walk, on certain classes of graphs. Our experiments show that state of the art models can already be enhanced with restricted quantum features that are classically efficient to compute. This study provides strong indications that the full leverage of quantum hardware can lead to development of high-performance architectures for certain tasks. In particular, Neutral Atom quantum hardware (Henriet et al., 2020) would be particularly suited to the type of time-dependent Hamiltonian we described here. Furthermore, we can create artificial classification tasks that are easily solvable with quantum enhanced models while classical models struggle. While the exact capabilities of our approach have to be explored, the results we obtain show that quantum enhanced GNNs are a promising family of models that could be fully exploited with near term quantum hardware.

## REPRODUCIBILITY STATEMENT

The code to reproduce the experiments discussed in the main text is included in the supplementary materials with indications to run it. We also detail in the appendices C.1 C.2 C.4 the details of the protocols.

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

# A METHODS AND THEORY

## A.1 LEARNING POSITIONAL ENCODINGS

Here we examine the responsiveness of the model presented section 3.2.2 to the initial values of $\theta, t$, and $\delta$. To do so, we compare the distributions of the $3 \times k$ parameters before and after training the transformer model. In our case, these initial values are directly related to the initial values of the $P_{\mathbf{W}}$ model parameters, as well as its architecture.

This comparison is summarized in figure 2. We can see on the top figures of fig.2 that the initial variances of $\theta, t$, and $\delta$ are small (0.03) which leads to a distribution of the values of the positional encoding that is highly peaked around zero (green curves) either for the diagonal elements (bottom left of fig.2) with a variance of a magnitude of $10^{-10}$ or the off diagonal elements (bottom right of fig.2) with a variance of a magnitude of $10^{-8}$.

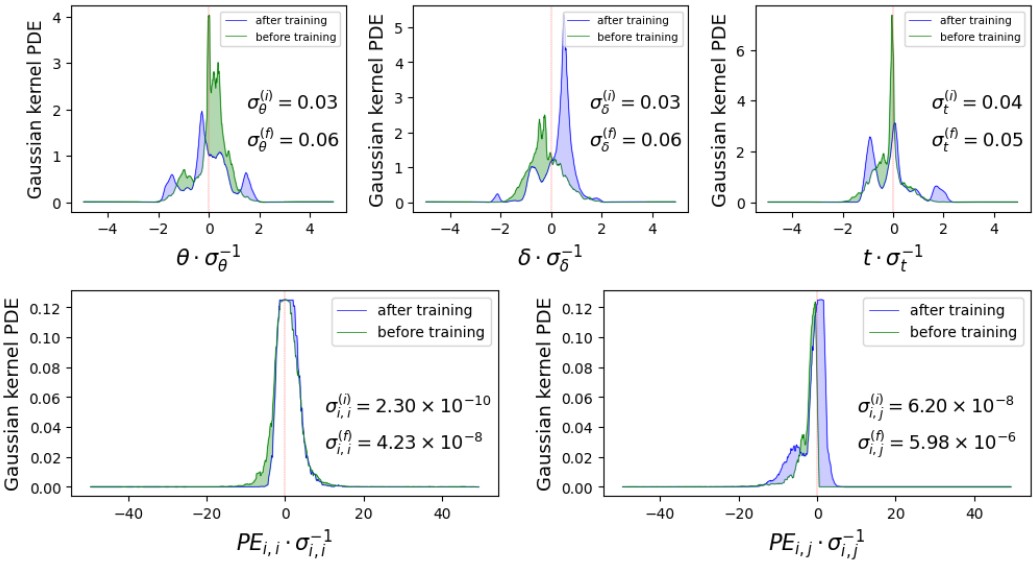

Figure 2: The probability density estimation (PDE) of the parameters output by the $P_{\mathbf{W}}$ model before (green curve) and after (blue curve) training. In the top figures we have the distributions of (from left to right) $\theta$, $\delta$ and $t$ normalized by their respective variances. The bottom figures show the distributions of the QCorr positional encoding (obtained from the Ising Hamiltonian) resulting from such parameters for the diagonal elements (bottom left figure) and off-diagonal elements (bottom right figure). The green curves correspond to the distribution of the PE obtained from the model before the training, and the blue curves those from the trained model. Here again, the distributions are normalized by their respective variances. All the PDEs are obtained using a Gaussian kernel.

Although these distribution become wider after the training of the model, since the variance doubles in each of these distributions as we can see it in all three of the top curves of fig.2, and even gains a factor of 100 in the distribution of the elements of the positional encoding (bottom curves in fig.2), it still results in very small values in the latter case. We can see from fig.3b that the gray band delimiting the variance around the mean value of each of the parameters $\theta$ (blue), $t$ (red) and $\delta$(green) includes only two regimes, and both are very close to zero, whereas for larger randomly extracted variances, we observe multiple regimes in which the positional encoding values display more diverse behavior fig.3a. In order to exploit this more diversified regime, we can either set a relative scaling between the output values of the $P_{\mathbf{W}}$ encoder, or fix its initial parameters appropriately, so that its outputs benefit from a higher variance. The former option is more compatible with a potential implementation of such approaches on real neutral-atom QPU hardware (Henriet et al., 2020), and requires further study, which relies on considerations from the quantum phase transitions of the Ising model (Boel & Kasteleyn, 1978; Dutta et al., 1996; Suzuki et al., 2013), to be properly addressed. It is not discussed in this paper and will be a a key aspect of future work.

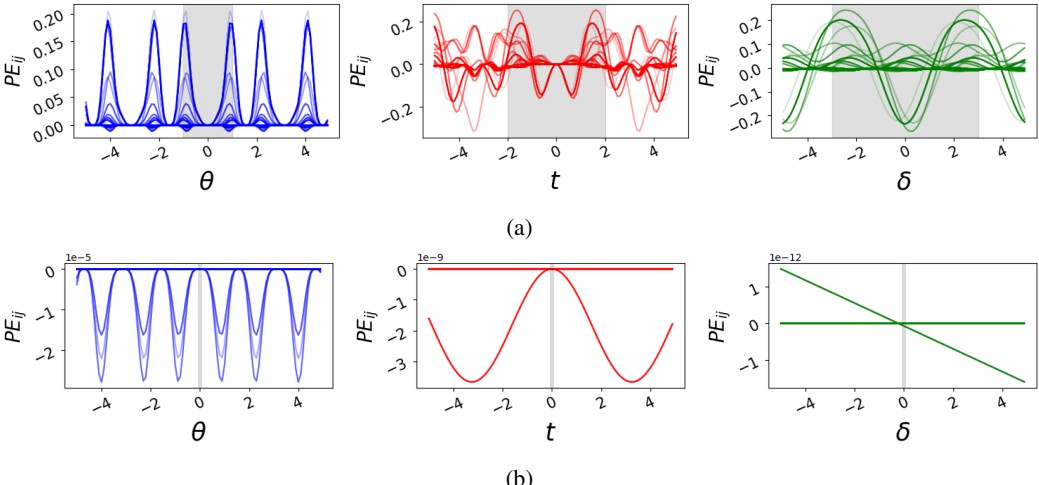

(a)

(b)

Figure 3: In **(a)** we have the values of the QCorr PE (all the values in the tensor are superposed in this figure) for a set of parameters $\theta, t$ and $\delta$ chosen randomly from a normal distribution of zero mean and fixed variance (of arbitrary values 1, 2 and 2.5). The blue curve shows the values of the PE when $\theta$ varies while $t$ and $\delta$ are fixed, the red curve is for varying $t$ and the green curves for varying $\delta$. The grey vertical tape indicates the variance of the normal distribution. In **(b)** we plot the same quantities, for the adjusted parameters after training. Here we used a 2 layered GAT model (Veličković et al., 2018) as a parameter encoder. In each case, we used a batch of 32 graphs from the Zinc (Gómez-Bombarelli et al., 2018) dataset to display these results

## A.2    DATA RANDOMIZATION

Here we describe the different randomization processes that we use in order to evaluate the contributions of the positional encoding and the external features in the model's results. Each process is summarized in figure 4

1. The first model is trained on a dataset where each graph is replaced by its configuration model (Newman, 2010). The graphs in this model are obtained by randomizing structural features, more precisely by arbitrarily rewiring pairs of edges (so-called double edge swapping), thus completely destroying the graph's structural information, with the exception of its degree sequence. A model trained on such data only receives relevant information from the external features associated with the dataset. An example of a single double edge swap is represented in figure 4a.

2. A second model is trained in an opposite way, by randomly permuting the elements of the external features vector, while keeping the structural information intact (no double edge swap is performed). A model trained in such data only receives relevant information from the positional encoding, here kept as the classical RRWP random walks tensor. An example of such randomization is given on figure 4c

3. In order to separate the contribution of the external features from that of the degree sequence (which is explicitly exploited in the GRIT model), we add one more structural randomization in which we do not keep the degree sequence anymore. The only structural feature conserved in this case is the graph density. To do so, we replace each graph in the dataset by a graph with a graph chosen randomly from the set $G_{n,m}$ of all graphs with the same number of nodes $n$ and the same number of edges $m$, thus removing the information related to the degree sequence. An example of such randomization if provided in figure 4b

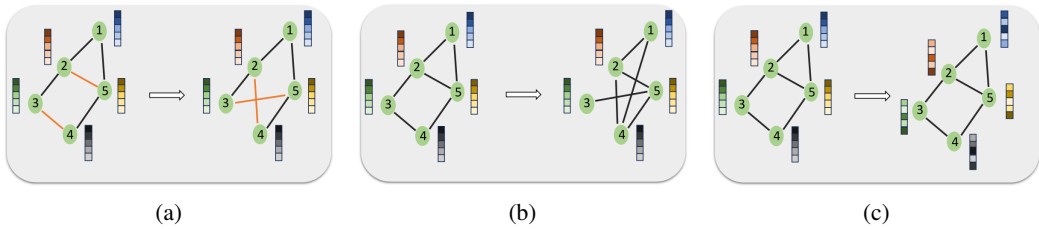

Figure 4: **(a)** An example of one double edge swap, the configuration model related to the input graph is obtained after a fixed (large) number of edge swaps. **(b)** a random graph picked from the class of $G_{n,m}$ graphs with $n$ nodes and $m$ edges (identical to those of the input graph). **(c)** a randomization that only concerns the external features (here represented at node level), in which each vector undergoes a random permutation of its elements.

### A.2.1 FORMAL EXPRESSION FOR THE CORRELATION MATRIX OF THE ISING HAMILTONIAN

Here we consider a specific case of the equation 2 , with $p = 1$, $\theta_0 = -\theta_1 = \theta$, $\hat{\mathcal{H}}_M = \sum_{i \in \mathcal{V}} Y_i$ and $\hat{\mathcal{H}}_\mathcal{G} = \sum_{(i,j) \in \mathcal{E}} Z_i Z_j - \delta \sum_{i \in \mathcal{V}} Z_i$.

$$|\psi_\mathcal{G}(\theta, t)\rangle = e^{i\theta \hat{\mathcal{H}}_M} e^{-it \hat{\mathcal{H}}_\mathcal{G}} e^{-i\theta \hat{\mathcal{H}}_M} |\psi_0\rangle ,$$

We then compute the covariance matrix of the number of occupation observable $\hat{n}_i = \frac{1}{2}(I + Z_i)$, obtained from $\langle n_i n_j \rangle - \langle n_i \rangle \langle n_j \rangle$, where $\langle n_i n_j \rangle = \langle \psi_\mathcal{G} | \hat{n}_i \hat{n}_j | \psi_\mathcal{G} \rangle$.

In order to emphasise the structure of the expression, we introduce

$$w_{ij}(\theta, t) = \left( \cos^2 \theta + \sin^2 \theta \, e^{iJ_{ij}t} \right) \tag{5}$$

and

$$\varrho_i^{(\delta)}(\theta, t) = e^{i\delta t} \prod_j w_{ij}(\theta, t), \tag{6}$$

Introducing

$$w_{ij}^\pm(\theta, t) = \prod_{l \in \mathcal{N}_{ij} \cup \{i,j\}} \cos^2 \theta + \sin^2 \theta \, e^{i(J_{il} \pm J_{jl})t}$$

the correlation between densities at $i$ and $j$ can then be expressed as

$$\langle n_i n_j(t) \rangle - \langle n_i(t) \rangle \langle n_j(t) \rangle = 4 \sin^4 \theta \cos^4 \theta \, \Re \left\{ \left[ \varrho_i^{(\delta)}(\theta, t) + \varrho_j^{(\delta)}(\theta, t) \right] \left[ 1 - w_{ij}(\theta, t)^{-1} \right] \right.$$
$$+ \frac{1}{2} \left[ 1 - e^{iJ_{ij}t} w_{ij}^+(\theta, t)^{-1} \right] \varrho_i^{(\delta)}(\theta, t) \varrho_j^{(\delta)}(\theta, t)$$
$$+ \frac{1}{2} \left[ 1 - w_{ij}^-(\theta, t)^{-1} \right] \varrho_i^{(\delta)}(\theta, t) \varrho_j^{(\delta)}(\theta, -t) \left. \right\} .$$

where $\varrho_j^{(\delta)}(\theta, -t)$ is equal to the complex conjugate of $\varrho_j^{(\delta)}(\theta, t)$. This is uniformly equal to zero if $i$ and $j$ are not neighbours and do not share any neighbours, and has peaks whenever $t = k \frac{\pi}{J_{il} \pm J_{jl}}$ or $t = k \frac{\pi}{J_{ij}}$ for $k \in \mathbb{Z}$.

## A.3 THEORY

### A.3.1 PROOF OF THEOREM 1

Strongly regular graphs Gamble et al., 2010 are graphs of $N$ vertices such that

- all vertices have the same degree $k$
- each pair of neighboring vertices has the same number of shared neighbors $\lambda$
- each pair of non-neighboring vertices has the same number of shared neighbors $\mu$

One can show (Shiau & Joynt, 2003; Gamble et al., 2010) that for strongly regular graphs, the powers of the adjacency matrix $A$ can be expressed as

$$A^n = \alpha_n I + \beta_n J + \gamma_n A$$

where $\alpha_n, \beta_n, \gamma_n$ only depend on $N, k, \lambda, \mu$. $I$ is the identity matrix, $J$ is the matrix full of 1s. The degree matrix is also equal to $kI$, then $(D^{-1}A)^n = A^n/k^n$.

Hence the information about distance contained in $\mathbf{P}_{uv}$ for strongly regular graphs is the same as in their adjacency matrices. Therefore, for strongly regular graphs, the GD-WL with RRWP test is equivalent to the WL test.

Furthermore, (Bodnar et al., 2021) proved that strongly regular graphs cannot be distinguished by the 3-WL test. Therefore GD-WL test with RRWP cannot distinguish non isomorphic strongly regular graphs.

We now show empirically that the GD-WL test with discrete 2-QiRW can distinguish between an example of two non-isomorphic strongly regular graphs with $N = 16, k = 6, \lambda = 2, \mu = 2$, example also taken by (Bodnar et al., 2021).

Those two graphs named $G1$ and $G2$ are shown figure 5, and the data are taken from (Spence). In order to do this, we calculate the distance matrix for the GD-WL test using the 2-QiRW encoding for an arbitrary $K > 2$. We compute the distance between node $i$ and node $j$,

$$\mathbf{D}_{ij} = \left[ \langle ij | (\hat{\mathcal{H}}_2^{XY})^K | \psi_{\text{init}} \rangle \, | k \in [0, K] \right]_{ij}$$

where $\psi_{\text{init}}$ is the uniform distribution over all edges of the graph. We perform the GD-WL test with $K = 5000$ and successfully distinguishes the two graphs.

We illustrate in figure 5 the distance between the encodings of the two graphs. We show the norm of the difference between the vectors consisting of the encoding flattened and sorted such that the measure is invariant with respect to the initial labeling. The code is available in the supplementary material to reproduce the experiment.

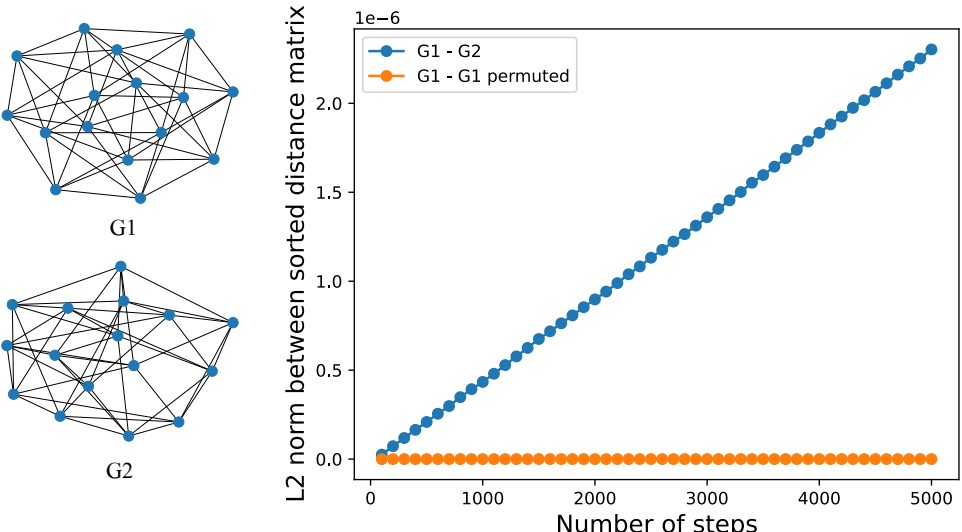

Figure 5: Two non-isomorphic strongly regular graphs G1 and G2 with $N = 16, k = 6, \lambda = 2, \mu = 2$ from (Bodnar et al., 2021). Difference between the discrete 2-QiRW encodings of G1 and G2. We additionally plot the difference between G1 and a permutation of labels of G1 to emphasize the difference.

### A.3.2 State graphs

Given a quantum system described by a hamiltonian $\hat{\mathcal{H}}$, acting on a Hilbert space $\mathbb{H}$, and a basis $\{|a_i\rangle\}_{i=1}^N$ of $\mathbb{H}$, we define a *state graphs*(Henry et al., 2021) $\mathcal{G}_{\hat{\mathcal{H}}} = (\mathcal{V}, \mathcal{E})$ in the following way :

- Each node $i$ in $\mathcal{V}$ is associated with a state $|a_i\rangle$.

- The set of edges $\mathcal{E}$ contains each pair of nodes $(i, j)$ such that $\langle j| \hat{\mathcal{H}} |i\rangle \neq 0$.

The dynamics of the quantum system can then be seen as a QRW on $\mathcal{G}_{\hat{\mathcal{H}}}$, where the (complex) hopping rate from node $i$ to node $j$ is given by $\langle j| \hat{\mathcal{H}} |i\rangle$. The $XY$ model preserves the number of excitations, therefore if one chooses $\{|a_i\rangle\} = \{|i_1 \dots i_k\rangle, (i_1, \dots, i_k) \in \{0,1\}^k\}$ as a basis, then the graph splits into $N+1$ non-overlapping subgraphs $\mathcal{G}_k$, each corresponding to a different number of particles $k$ (or hamming weights), as illustrated in figure 6. If the system is initiated in a state with $k$ paticles, then its dynamics will be restricted to $\mathcal{G}_k$.

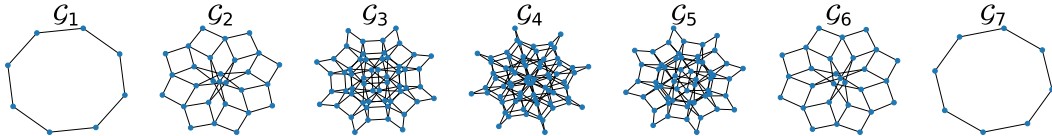

Figure 6: State graphs for a $XY$ model on an arbitrary graph of 8 nodes. $\mathcal{G}_i$ contains all states with $i$ particles (figure taken from (Henry et al., 2021)).

## B   Graph Transformer with Quantum Correlations

This section presents one of our proposal GTQC (Graph Transformer with Quantum Correlations), an architecture of Graph Neural Network based on Graph Transformers and incorporating global graph features computed with quantum dynamics. A global view of the algorithm is represented on figure 7. Representation learning on graphs using neural network has become the state of the art of graph machine learning (Wu et al., 2020). Scaling deep learning models has brought lots of benefits as shown by the success of large language models (Brown et al., 2020; Alayrac et al., 2022). The goal was to bring the best of both worlds, meaning large overparameterized deep learning models, and structural graph features intractable with a classical computer. The architecture we propose only uses nodes features, but similar techniques could be implemented for edges features.

### B.1   Transition Matrix from Quantum Correlations

Here we develop a method to compute a parameterized transition matrix or quantum *attention* matrix from the correlations of a quantum dynamic. It is done with a quantum computer, or Quantum Processing Unit (QPU). This matrix will later be used in the update mechanism of our architecture. Once the quantum attention matrix is computed, the rest of the architecture is purely classical, and all existing classical variations could be implemented. Finally, the quantum attention matrix is by construction equivariant to a permutation of the nodes.

We consider a parameterized graph state as defined in equation 2, parameterized by the trainable parameter $\boldsymbol{\theta} = (\theta_0, t_0, \theta_1, t_1, \dots \theta_p)$, and noted $|\psi(\theta)\rangle$. $\boldsymbol{\theta}$ will be called the *quantum parameters* in the rest of the section. We then compute for every pair of nodes $(i, j)$ the vector of 2-bodies observables $C_{ij} = [\langle Z_i Z_j\rangle, \langle X_i X_j\rangle, \langle Y_i Y_j\rangle, \langle X_i Z_j\rangle, \langle X_i Y_j\rangle, \langle Y_i Z_j\rangle, \langle X_j Z_i\rangle, \langle X_j Y_i\rangle, \langle Y_j Z_i\rangle]^T$ where $\langle O \rangle = \langle\psi(\theta)| O |\psi(\theta)\rangle$.

The quantum attention matrix is computed by taking a linear combination of the previous correlation vector and optionally a softmax over the lines.

$$A(\theta)_{ij} = \gamma^T C_{ij} \tag{7}$$

$$A(\theta)_{ij} = softmax(\gamma^T C_{ij}) \tag{8}$$

where $\gamma$ is a trainable vector of size 9. Multiple correlators are measured to enrich the number of features that can be extracted out of the quantum state. Limiting ourselves to e.g $\langle Z_i Z_j\rangle$ might be

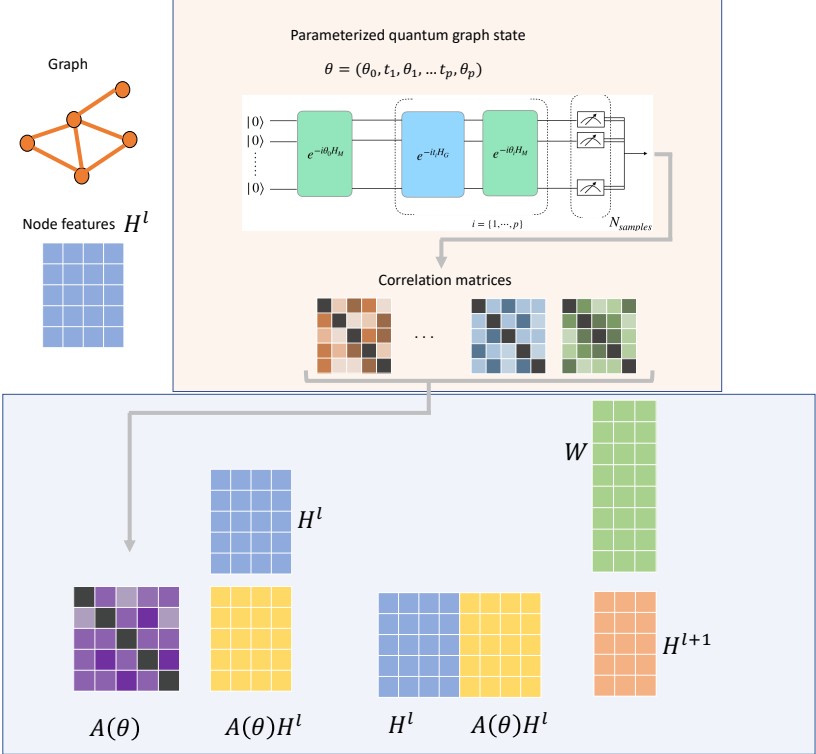

Figure 7: Overview of our model. The graph is translated into a hamitonian that will drive a quantum system. After a parameterized evolution, the correlations are measured and aggregated into a single attention matrix.

inefficient because $\langle \psi_f | Z_i Z_j | \psi_f \rangle$ could be written $XX^\dagger$ where $X$ is a matrix with row $i$ equal to $(Z_i | \psi_f \rangle)^\dagger$. The resulted weight matrix is therefore a symmetric positive semi-definite matrix, and can be reduced by a Choleski decomposition $A = LL^T$ where $L$ is a real matrix of shape $N \times N$. The same model can then be constructed by learning the matrix L even though it is unclear if that would be efficient.

## B.2 UPDATE MECHANISM

The quantum weight matrix previously computed is used as a transition matrix, or attention matrix, in the update mechanism of our model. Given $H^l$ the node features matrix at the layer $l$, the node features matrix at the next layer is computed with the following formula :

$$H^{l+1} = \sigma((A(\theta)H^l || H^l)W) \qquad (9)$$

where $\sigma$ is a non-linearity, $H$ is of size $(N \times d)$ where each row represents a node feature of dimension $d$, $W$ is a learnable weight matrix of size $(2d \times d_h)$, $A(\theta)$ is the attention matrix with parameters $\theta$ computed in B.1 and $||$ is the concatenation on the columns. With the same approach as Transformers or Graph Attention Networks, one can use several attention heads per layer to improve the expressivity of the architectures. The update formula is given by $H^{l+1} = ||_i^{N_{heads}} HEAD_i(H^l)$.

where each head is computed with the formula B.2. The total dimension of the feature vector is $N_{heads}d_h$. Each head has a different quantum circuit attached, and can be computed in parallel if one possesses several QPUs.

### B.3 EXECUTION ON A QUANTUM DEVICE

We provide here some precisions on how our model would be implemented on quantum devices.

**Decoupling between quantum and classical parts**
The parameters of the quantum states and the classical weight parameters are independent in our algorithm. One can then asynchronously measure all the quantum states of the model and run the classical part. This may be particularly important for NISQ implementation since the access of QPUs are quite restricted in time. Furthermore, the gradients of the classical parameters depend only on the correlation matrices, so they can be easily computed with backpropagation without any supplementary circuit run.

**Training the parameters of the quantum state**
Computing the gradients of parameterized quantum circuits is a challenge source of numerous research in the quantum computing community (Kyriienko & Elfving, 2021; Wierichs et al., 2022; Banchi & Crooks, 2021). Finite-difference methods fail because of the sampling noise of quantum measurements and the hardware noise. Some algorithms named parameter-shift rules were then created to circumvent this issue (Mitarai et al., 2018). In some cases, the derivative of a parameterized quantum state can be expressed as an exact difference between two other quantum states with the same architecture and other values of the parameters.

We detail here how we would compute the gradient in a simple case of our architecture. Let $\hat{\mathcal{H}}$ be a hamiltonian, $\hat{\mathcal{O}}$ an observable, $|\psi_0\rangle$ an initial state. We introduce

$$|\psi(\theta)\rangle = U(\theta)|\psi_0\rangle = \exp(-i\theta\hat{\mathcal{H}})|\psi_0\rangle \tag{10}$$

$$f(\theta) = \langle\psi(\theta)|\,\hat{\mathcal{O}}\,|\psi(\theta)\rangle \tag{11}$$

It is known (Wierichs et al., 2022) that $f$ can be expressed as a trigonometric polynomial

$$f(\theta) = \sum_{\omega\in\Omega} a_\omega \sin(\omega\theta) + b_\omega \cos(\omega\theta) \tag{12}$$

where $\Omega$ is a finite set of frequencies whose values are equal to the gaps between eigenvalues of $\hat{\mathcal{H}}$. In the case of $\hat{\mathcal{H}} = \hat{\mathcal{H}}_M$, the frequencies are the integers between 0 and $N$. One can then evaluate $f$ on $2N + 1$ points and solve the linear equations to determine $\{a_\omega\}$, $\{b_\omega\}$ and the derivative of $f$. This is the same for the $\hat{\mathcal{H}}^I$ Hamiltonian which associated frequencies are the integers between 0 and $|\mathcal{E}|$.

**Alternate training**
At the time of this work, quantum resources are very expensive, so we want to limit ourselves in the number of access to the QPU. One way to do so is not to update every parameter at each epoch. Typically the gradients of the quantum parameters are expensive to compute so the update would be less frequent. Due to the decoupling between the quantum and classical parameters, one is able to compute the gradients of the weights matrix with only the quantum attention matrices stored in memory.

**Random parameters**
Optimizing over the quantum parameters can be costly and ineffective with current quantum hardware. Even with emulation, back-propagating the loss through a system of more than 20 qubits is very difficult. We encounter memory errors for more than 21 qubits on our A100 GPUs, even though our implementation is certainly not optimal. Therefore we propose an alternative scheme to our model, to help with both actual hardware implementations and classical emulation.

The main idea in the spirit of (Rahimi & Recht, 2008) is to evaluate the attention matrices on many random quantum parameters, and only training the classical weights. From a model $f(x; W, \theta) = \left\|\right._i^{N_{heads}} \sigma((A(\theta_i)H^l||H^l)W_i)$ with one layer, we would normally find the parameters that minimize a loss between inputs $x$ and labels $y$

$$W^*, \theta^* = \arg\max_{W,\theta} \sum_{i=1}^{M} l(f(x_i; W, \theta), y_i) \tag{13}$$

Instead, we create a model with more heads and fixed random values. $\theta'$ expressed as $f(x; W, \theta') = \left\|\right\|_i^{N'_{heads}} \sigma((A(\theta'_i)H^l\|H^l)W_i)$ and we minimize only on $\theta$

$$W^* = \arg\max_W \sum_{i=1}^{M} l(f(x_i; W, \theta'), y_i) \tag{14}$$

### B.4 SPECTRAL VERSION OF OUR MODEL

The spectral version of our model consists of formally identify the correlation matrix $C$ with the laplacian matrix in the formulas of the previous sections. This replacement is possible because the correlation matrix has the key properties to be symmetric and positive semi-definite just like the Laplacian, so all the expressions have a proper definition. If we note $C = V_c \Lambda_c V_c^T$ with $V_c$ orthogonal and $\Lambda_c$ diagonal, we can define pseudo-Fourier transform, pseudo-convolution and pseudo-filtering operation by

$$\tilde{f} = V_c^\top f \tag{15}$$

$$f * g = V_c^\top ((V_c f) \odot (V_c g)) \tag{16}$$

$$g = \gamma_\theta(C)f = \gamma_\theta(V_c \Lambda_c V_c^\top)f = V_c \gamma_\theta(\Lambda_c)V_c^\top f \tag{17}$$

These definitions come from (Shuman et al., 2013; Bruna et al., 2013; Defferrard et al., 2016). The analogy we drew between the correlation matrix and the Laplacian is only a formal one, hence the term "pseudo-Fourier". Indeed, the correlation matrix has not the same characteristics as the Laplacian. People came up with the previous formalism in order to generalize signal processing to arbitrary graphs (Shuman et al., 2013). Convolution operators were well defined for continuous spaces or images, and convolutional neural networks showed a remarkable power in computer vision tasks. It was therefore natural to look for an extension to a broader family of spaces. In the same way that a grid can be viewed as a discretization of a plane, the interpretation of the framework is natural considering that an arbitrary graph can be viewed as the discretization of an arbitrary surface.

We have that the powers of the Laplacian correspond to the nodes that are connected, whereas this is not true for the correlation matrix.

## C EXPERIMENTS

### C.1 EXPERIMENTS ON QUANTUM RANDOM WALK

In the same way as (Ma et al., 2023), we perform the experiments on the standard train/val/test splits. For each dataset, we perform 4 runs with the seeds 0, 1, 2, 3 and display the average of the scores and the standard deviation.

We do not perform an extensive hyperparameter search, and we only compute ourselves the GRIT model. We take the same hyperparameters as (Ma et al., 2023) that we remind in table 4.

### C.2 EXPERIMENTS ON SYNTHETIC DATASETS

We benchmark GPS and GCN models with different types of positional encoding, the random walk, laplacian eigenvectors, and the eigenvector of quantum correlations. We try all combinations of hyperparameters among the following

- dimension of position encoding : 10, 20, 50
- number of layers : 2, 4
- hidden dimension : 32, 64, 128
- type of layer : GPS, GCN

We also try the GRIT model with RRWP with the same hyperparameters as for ZINC. The biggest models have around 300k parameters, whereas the smallest ones have around 20k. We train for 200

Table 4: Hyperparameters for five datasets from BenchmarkingGNNs (Dwivedi et al., 2020), ZINC-full (Irwin et al., 2012) and (Hu et al., 2021)

| Hyperparameter | ZINC/ZINC-full | MNIST | CIFAR10 | PATTERN | CLUSTER | PCQM4Mv2 |
|---|---|---|---|---|---|---|
| # Transformer Layers | 10 | 3 | 3 | 10 | 16 | 16 |
| Hidden dim | 64 | 52 | 52 | 64 | 48 | 256 |
| # Heads | 8 | 4 | 4 | 8 | 8 | 8 |
| Dropout | 0 | 0 | 0 | 0 | 0.01 | 0.1 |
| Attention dropout | 0.2 | 0.5 | 0.5 | 0.2 | 0.5 | 0.1 |
| Graph pooling | sum | mean | mean | − | − | mean |
| PE dim (RW-steps) | 21 | 18 | 18 | 21 | 32 | 16 |
| PE encoder | linear | linear | linear | linear | linear | linear |
| QPE dim (1CQRW steps) | 20 | 18 | 18 | 20 | 32 | 16 |
| Max duration | $\pi$ | $\pi$ | $\pi$ | $\pi$ | $\pi$ | $\pi$ |
| Min duration | 0.1 | 0.1 | 0.1 | 0.1 | 0.1 | 0.1 |
| Initial distribution | local | local | local | local | local | local |
| QPE dim (2QiRW steps) | 20 | 18 | 18 | 20 | 32 | 16 |
| Initial distribution | adjacency | adjacency | adjacency | adjacency | adjacency | adjacency |
| Batch size | 32/256 | 16 | 16 | 32 | 16 | 256 |
| Learning Rate | 0.001 | 0.001 | 0.001 | 0.0005 | 0.0005 | 0.0002 |
| # Epochs | 2000 | 200 | 200 | 100 | 100 | 150 |
| # Warmup epochs | 50 | 5 | 5 | 5 | 5 | 10 |
| Weight decay | $1e-5$ | $1e-5$ | $1e-5$ | $1e-5$ | $1e-5$ | 0 |
| # Parameters GRIT | 473,473 | 102,138 | 99486 | 477,953 | 432,206 | 11.8M |
| # Parameters 2QiRW GRIT | 476,033 | 104,010 | 101,358 | 480,513 | 434,742 | 11.8M |

epochs using the Adam optimizer, 0.001 learning rate, no weight decay. We split randomly the dataset on train/validation/test with a proportion 0.8/0.1/0.1, and we measure the test accuracy of the model having the highest validation accuracy.

We illustrate figure 8 the fact that quantum encodings are distinctive features of the two categories of the S-PATTERN dataset whereas classical encodings are not.

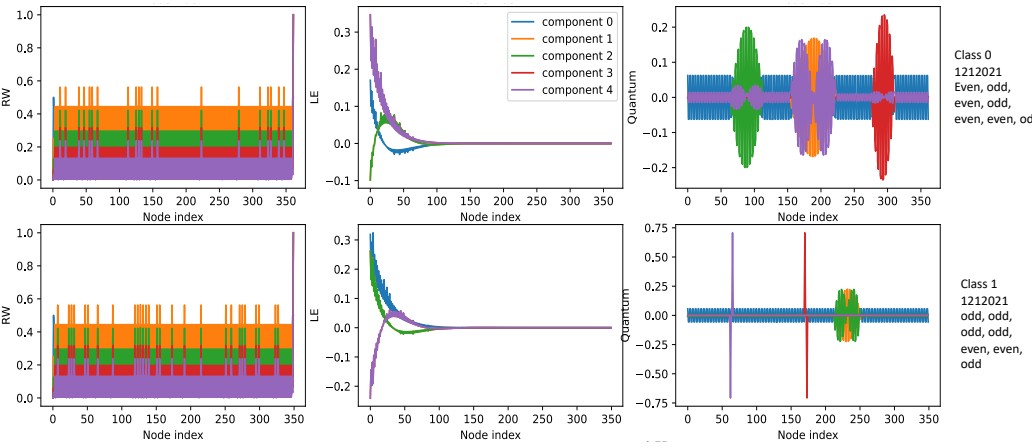

Figure 8: Different positional encodings for graphs of each class of the C-LADDER dataset. RW: Random Walk Structural Encodings. LE: Laplacian Eigenvectors. Quantum: Eigenvectors of the correlation on the ground state. The quantum encodings are very distinctive of each class, which is not the case for the other ones.

## C.3 EXPERIMENTS ON GROUND STATE EIGENVECTORS

In this subsection, we experiment the use of the eigenvectors of the correlation matrix on the ground state as node features in comparison to other PE such as the laplacian eigenvectors or random walk encodings. We experiment two architectures (SAGE, GPS) and benchmark our encodings on three standard chemistry datasets (QM7, QM9, ZINC) for regression tasks. All results are available in

table 5. We show the scores for a sample of hyperparameters of the models, and we include cases where our model perform either worse or better than other approaches. The use of quantum features achieves the best overall score for QM7, whereas a mix of LE and RW performs the best for QM9, and we are able to retrieve the fact from (Rampášek et al., 2022) that the RW features perform the best on ZINC.

We don't reach the same score because we didn't include the edge features. For QM9, the features are rescaled such that their average across the dataset is 0 and their variance is 1. The quantum features are computed by brute forcing the ground state of all graphs until size 30. Only ZINC has some graphs whose size is between 31 and 38, in that case the quantum features are set to 0. For ZINC we do not include the edge features.

Table 5: Results of GPS and other GNNs with different input features on real datasets. We use the Adam estimator for 500 epochs and a learning rate of .001. For ZINC, standard partitions train/val/test are given. For QM7, QM9 we report the average error over 5 random partitions train/val/test of proportion .8/.1/.1. We take 10-dimensional features for each type. The hyperparameters are (#heads, #layers, #hidden neurons).

| Dataset | QM9 | | | QM7 | | | ZINC | |
|---|---|---|---|---|---|---|---|---|
| Model | GPS | SAGE | GPS | GPS | SAGE | GPS | GPS | GPS |
| Hyperparameters | (1, 2, 128) | (1, 2, 128) | (4, 3, 256) | (1, 2, 256) | (1, 2, 128) | (4, 2, 128) | (4, 5, 32) | (8, 10, 64) |
| No PE | 2.96 | 3.28 | 1.75 | 32.23 | 60.45 | 34.97 | 0.53 | 0.43 |
| LE | 1.95 | 2.19 | 1.56 | 14.68 | 36.62 | 15.40 | 0.67 | 0.65 |
| RW | 2.28 | 2.93 | 1.47 | 16.18 | 14.97 | 16.45 | **0.29** | **0.27** |
| LE+RW | 1.83 | 2.05 | **1.42** | **14.48** | 14.27 | 14.81 | 0.44 | 0.40 |
| Q (ours) | 2.13 | 2.7 | 1.79 | 16.15 | 15.85 | 15.36 | 0.67 | 0.66 |
| LE+RW+Q (ours) | **1.80** | **2.02** | 1.49 | 14.66 | **14.13** | **14.03** | 0.56 | 0.55 |

## C.4 EXPERIMENTS ON THE MODEL DESCRIBED IN APPENDIX B

### C.4.1 TRAINING ON GRAPH COVERS DATASET

We test the performances of our model on a dataset of non isomorphic graphs constructed to be indistinguishable by the WL test. We believe that this task will be especially difficult for classical GNNs and could constitute an interesting benchmark. The way to construct such graphs has been investigated in (Bamberger, 2022) and more details can be found in the Appendix D.1. We compare the training loss with one of the most recent implementation of graph transformers (GraphGPS) by (Rampášek et al., 2022). The results and implementation details are in figure 9. We can see that our model is able to reach much lower values of the loss than GraphGPS, even if both model acheive 100% accuracy after 20 epochs.

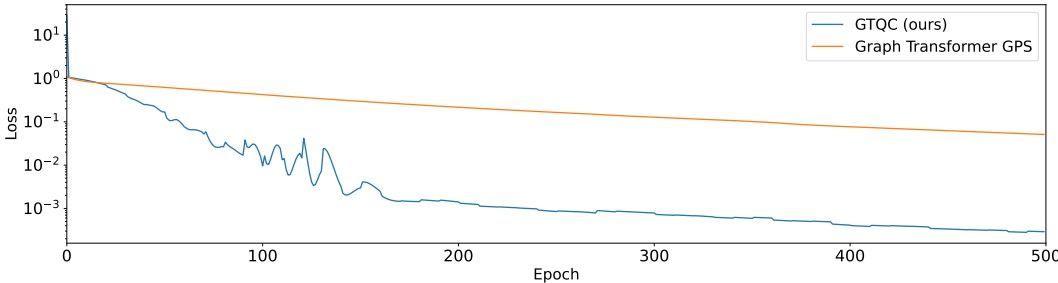

Figure 9: Training loss for our model and a recent graph transformer implementation (GraphGPS) (Rampášek et al., 2022). We both train our model and the GraphGPS model with two layers, a hidden dimension of 128, and 10 dimensional position embedding for the spectral attention of GraphGPS. We use a learning rate of 0.1 and no weight decay. Both models achieve 100% accuracy after 20 epochs.

### C.4.2 BENCHMARK ON GRAPH CLASSIFICATION AND REGRESSION TASKS

We benchmarked our model GTQC and its randomized version of it with different GNN architectures. We selected different datasets from various topics and with diverse tasks to show the general capabili-

ties of our approach. We limited the size of the graphs to 20 nodes in order to be able to simulate the quantum dynamics, and performing a backpropagation. We then chose datasets with the majority of graphs falling below this size limit. The details of each dataset can be found in Appendix D.1.

We implemented the models with a classical emulator of quantum circuits implemented in *pytorch* (Paszke et al., 2019) and with *dgl* (Wang et al., 2019), and ran them on A100 GPUs. All experiments were done using one GPU, except QM9 which required 4. We used a Adam optimizer with a learning rate .001, no weight decay for 500 epochs. The quantum parameters were only updated every 10 epochs because of computation time. As an order of magnitude, one epoch of QM7 takes 6 min with 1 GPU, and one epoch of QM9 takes 1h with 4 GPUs. Most of the time is allocated to compute the quantum dynamics, the size of classical parameters has little effect on the compute time except for the 2048 hidden layers.

We compare our model to three architectures of message passing models : GCN (Kipf & Welling, 2016), SAGE (Hamilton et al., 2018), GAT (Veličković et al., 2018). In order to have a fair comparison between the models, we employ similar hyperparameters for all of them. We use ReLU function for all activation functions. Each model has 2 layers and 1 head for multi-head ones like ours and GAT, except on the randomized version of our model. We don't use apply softmax as described in section B.1, except for the randomized instance because we observed the training was more stable. We also report the results by comparing models that have the same number of hidden neurons per layer and therefore approximately the same number of parameters. Each dataset is randomly split in train, validation, test with respective ratios of .8, .1, .1 and the models are run on 5 seeds, except QM9 for which we have only one seed.

All metrics are such that the lower the better, for classification tasks we display the ratio of misclassified objects. For QM9 the loss is aggregated over all the targets. Figure 10 shows results as box plots accompanied by the underlying points. Though the variability of the results is a bit higher, GTQC reaches similar results than usual classical well-known approaches on QM7 and DBLP_v1 (even beating GCN on this dataset) and seems relevant on QM9 as well. GTQC random usually outperforms GTQC, illustrating the complexity of the optimisation of the quantum system. GTQC random provides very promising results on DBLP_v1 and outperforms all other approaches on QM9. Letter-med seems to represent a difficult task for our quantum methods as they both perform very poorly and way worse than classical methods. QM7 also seems to be challenging for GTQC random. Table 6 shows the results.

Table 6: Scores for each model

| model | dataset breadth | DBLP | Letter-med | QM7 | QM9 |
|---|---|---|---|---|---|
| GAT | 128 | 0.08 ± 0.00 | 0.22 ± 0.02 | 64.27 ± 6.40 | 4.03 ± 0.00 |
| | 512 | 0.08 ± 0.01 | 0.20 ± 0.03 | 60.42 ± 5.56 | 3.23 ± 0.00 |
| | 1024 | 0.08 ± 0.01 | 0.21 ± 0.04 | 56.06 ± 2.77 | 0.00 ± 0.00 |
| | 2048 | 0.08 ± 0.00 | 0.19 ± 0.04 | 59.27 ± 3.33 | 0.00 ± 0.00 |
| GCN | 128 | 0.09 ± 0.01 | 0.21 ± 0.05 | 67.50 ± 4.70 | 4.22 ± 0.00 |
| | 512 | 0.09 ± 0.01 | 0.16 ± 0.02 | 63.30 ± 2.67 | 3.44 ± 0.00 |
| | 1024 | 0.09 ± 0.01 | 0.15 ± 0.02 | 62.94 ± 2.90 | 0.00 ± 0.00 |
| | 2048 | 0.09 ± 0.01 | 0.14 ± 0.02 | 60.70 ± 2.30 | 0.00 ± 0.00 |
| GTQC | 128 | 0.08 ± 0.00 | 0.71 ± 0.06 | 68.95 ± 8.50 | 8.01 ± 0.00 |
| | 512 | 0.08 ± 0.00 | 0.66 ± 0.04 | 70.09 ± 18.48 | 6.23 ± 0.00 |
| | 1024 | 0.08 ± 0.01 | 0.67 ± 0.06 | 67.85 ± 16.78 | 5.27 ± 0.00 |
| | 2048 | 0.08 ± 0.00 | 0.73 ± 0.06 | 65.53 ± 2.48 | 5.30 ± 0.00 |
| GTQC random | 32 | 0.00 ± 0.00 | 0.56 ± 0.00 | 0.00 ± 0.00 | 0.00 ± 0.00 |
| | 64 | 0.00 ± 0.00 | 0.48 ± 0.00 | 0.00 ± 0.00 | 0.00 ± 0.00 |
| | 128 | 0.08 ± 0.01 | 0.49 ± 0.00 | 92.62 ± 2.82 | 2.34 ± 0.00 |
| SAGE | 128 | 0.08 ± 0.01 | 0.09 ± 0.02 | 62.30 ± 3.47 | 3.26 ± 0.00 |
| | 512 | 0.07 ± 0.00 | 0.09 ± 0.02 | 61.47 ± 3.06 | 2.41 ± 0.00 |
| | 1024 | 0.07 ± 0.00 | 0.09 ± 0.01 | 61.30 ± 3.07 | 2.23 ± 0.00 |
| | 2048 | 0.08 ± 0.01 | 0.09 ± 0.02 | 54.41 ± 6.42 | 0.00 ± 0.00 |

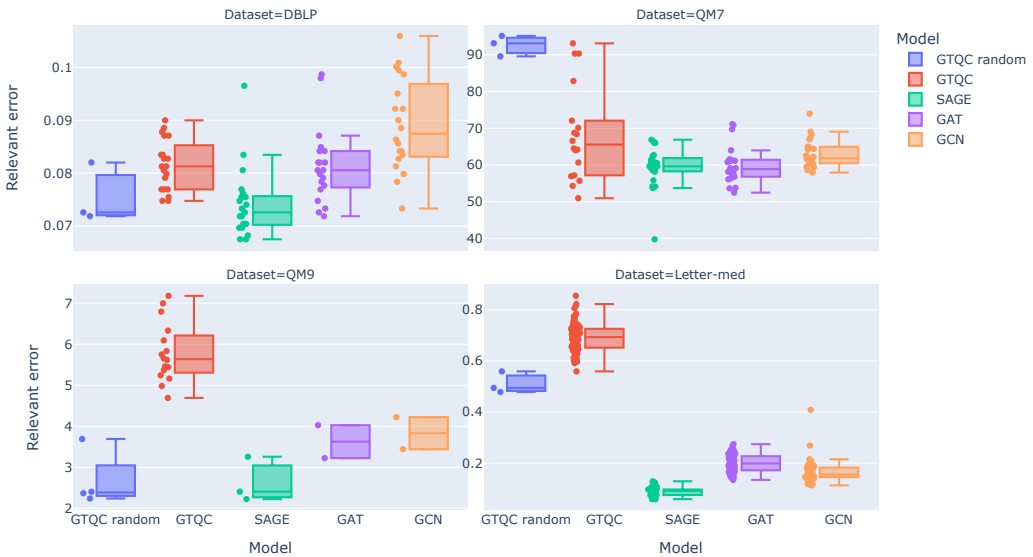

Figure 10: Summary graph of the results of the different models on the studied datasets. Each point is an instance of the model with specific hyperparameters and specific seed for dataset splits.

# D   SUPPLEMENTARY INFORMATION ABOUT THE DATASETS

## D.1   DATASETS USED IN EXPERIMENTS ON QUANTUM RANDOM WALKS

The datasets used for benchmarking the use of quantum random walks encodings are standard in the GNN community. The first five are from (Dwivedi et al., 2020), the last one is from (Hu et al., 2021). We reproduce the table of statistics 7 taken from (Ma et al., 2023), and we also refer the reader to (Rampášek et al., 2022) for more information about the datasets.

Table 7: Overview of the graph learning datasets involved in this work (Dwivedi et al., 2020), (Irwin et al., 2012), (Hu et al., 2021) .

| Dataset | # Graphs | Avg. # nodes | Avg. # edges | Directed | Prediction level | Prediction task | Metric |
|---------|----------|--------------|--------------|----------|------------------|-----------------|--------|
| ZINC(-full) | 12,000 (250,000) | 23.2 | 24.9 | No | graph | regression | Mean Abs. Error |
| MNIST | 70,000 | 70.6 | 564.5 | Yes | graph | 10-class classif. | Accuracy |
| CIFAR10 | 60,000 | 117.6 | 941.1 | Yes | graph | 10-class classif. | Accuracy |
| PATTERN | 14,000 | 118.9 | 3,039.3 | No | inductive node | binary classif. | Weighted Accuracy |
| CLUSTER | 12,000 | 117.2 | 2,150.9 | No | inductive node | 6-class classif. | Accuracy |
| PCQM4Mv2 | 3,746,620 | 14.1 | 14.6 | No | graph | regression | Mean Abs. Error |

### D.1.1   S-PATTERN

We define *strongly correlated graphs* as graphs that possesses only two ground states, and one state can be obtained by flipping all the bits from the other. The correlation matrix on this graph can be decomposed in four quadrants of 1s and -1s given a suitable permutation of the vertices. Examples of strongly correlated subgraphs are graphs composed of even cycles tiled next to each other. Importantly all strongly correlated graphs of same number of nodes have the same correlation matrix upon a suitable permutation of the nodes. However they may be extremely different by other characteristics like laplacian eigenvectors or random walk features. In our example, we will look to exploit this invariance.

Our example consists of strongly correlated subgraphs linked together by a small graph composed of triangles. It is illustrated in figure 11. The values of the correlations are diferrent for each class. For

one class, the absolute value of correlations between the two strongly correlated graphs is equal to 1/9 whereas for the other class it is equal to 1/11. The numerical difference is too small to be exploited with a traditional attention approach as explained in section **??**, but there is a striking difference when one computes the eigenvectors of the correlation matrices.

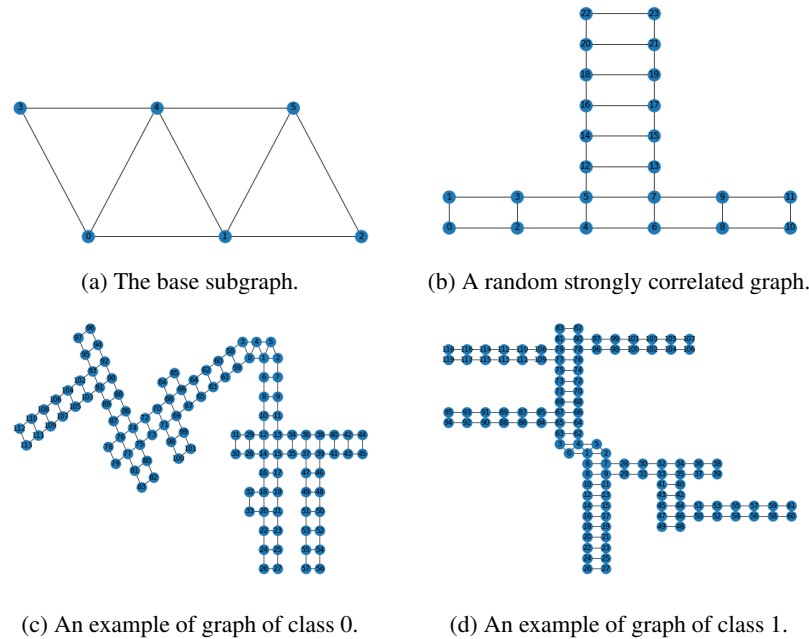

(a) The base subgraph.

(b) A random strongly correlated graph.

(c) An example of graph of class 0.

(d) An example of graph of class 1.

Figure 11: Construction of the S-PATTERN dataset. Two random strongly correlated graphs (11b) are linked to the base subgraph (11a), either in opposite sides, or in adjacent sides. In the class 0, the correlations between one strongly correlated graph to the other are in absolute value 1/9, whereas for class 1 this correlation is 1/11.

Our dataset consists of 1000 graphs per class, so 2000 in total, and is composed of graphs of about 450 nodes.

### D.1.2  C-LADDER

In this subsection, we explain how to construct the C-LADDER dataset. Our building blocks are 3 types of graphs, called types 0, 1, 2. Each type is composed of one ladder graphs with crossings inserted at different places. All crossings are in the same fixed arbitrary direction. Type 0 graphs are plain ladder graphs and their Ising hamiltonian has two ground states. Type 1 graphs are type 0 graphs with crossings separated with an odd number of nodes. The crossings are located such that they have one possible Ising ground state which is one of the ground states of the type 0 associated graph. The crossings will effectively select one of the two possible ground states. Type 2 graphs are ladder graphs of odd length with crossings at the beginning and the end. An illustration of the types of graphs is provided figure 12a.

We construct a graph given a sequence of types of graphs by concatenating a graph within each building block. The concatenation is made by adding edges to continue the ladder, the process is illustrated figure 12b. The ground state of the total graph is included in a union of the groundstate of the subgraph, so it can be efficiently computed for short sequence length (less than 10 subgraphs). The number of ground states of the whole graph will not change with the length of the subgraphs of type 0 and 1, but will depend on their parity. The length of type 2 graphs is necessarily even and the number of possible ground states grows linearly with the length. For a type 1 graph, the number of crossings or their exact location does not change the ground state.

Our dataset is composed of two categories of graphs, each one with the same sequence of types of graphs (1, 2, 1, 2, 0, 2, 1), with different parities of lenght of graphs. The parity for class 0 was (even, odd, even, odd, even, even, odd) whereas for class 1 it was (odd, odd, odd, odd, even, even, odd

). The length of graphs are chosen randomly in a uniform way between 10 and 52, the number of crossings in type 1 graphs are chosen randomly in a uniform way between 2 and 9.

Our dataset consists of 1000 graphs per class, so 2000 in total, and is composed of graphs of about 350 nodes.

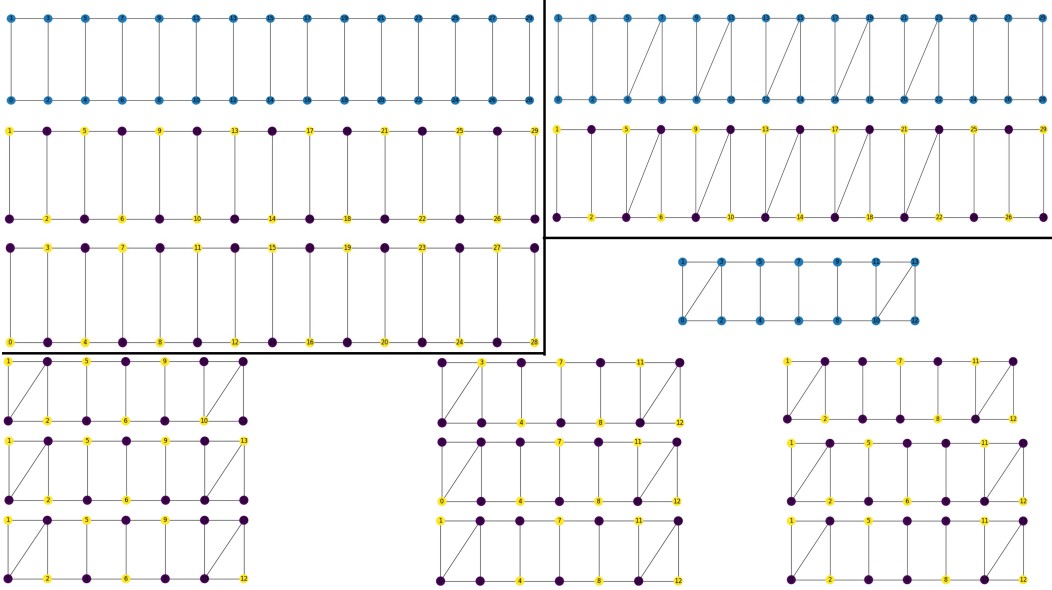

(a) The base subgraphs (type 0, type 1, type 2) and their possible ground state. Top left: type 0 graph of length 15, 2 possible ground states. Top right: type 1 graph of length 15, 1 possible ground state. Bottom : type 2 graph of length 9, 9 possible ground states.

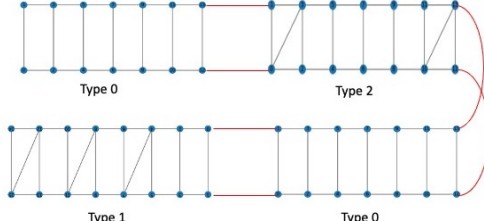

(b) Example of graph associated with sequence 0201. In red are the added edges.

Figure 12: Construction of our C-LADDER dataset

## D.2 QM7 AND QM9 MOLECULES AND GRAPH REGRESSION

**Context**
QM7 dataset is a subset of the GDB-13 database (Blum & Reymond, 2009), a database of nearly 1 billion stable and synthetically accessible organic molecules, containing up to seven heavy atoms (C, N, O, S). Similarly QM9 is a subset of the GDB-17 database consisting of molecules with up to nine heavy atoms. Learning methods using QM7 and QM9 are predicting the molecules electronic properties given stable conformational coordinates.

**QM7 figures**
QM7 consists of 7165 molecule graphs. Each node is an atom with its 3D coordinates and atomic number Z. The only edge feature is the entry of the Coulomb matrix. Each graph is thus fully connected and has one regression target corresponding to its atomization energy.

**QM9 figures**
QM9 consists of 130831 molecule graphs of between 1 and 29 nodes with an average of 18 nodes (see Figure 13). Each node is an atom with its 3D coordinates and its atomic number Z. Edges are

| Conf. category | Conferences | #Papers | #Graphs |
|---|---|---|---|
| DBDM | SIGMOD, VLDB, ICDE, EDBT, PODS, DASFAA, SSDBM, CIKM, DEXA, KDD, ICDM, SDM, PKDD, PAKDD | 20601 | 9530 |
| CVPR | ICCV, CVPR, ECCV, ICPR, ICIP, ACM Multimedia, ICME | 18366 | 9926 |

Table 8: DBLP_v1 details.

purely distance based and have no feature. Each graph is thus fully connected and has 12 regression targets corresponding to diverse chemical electronic properties. In our implementation, all the targets are recentered and rescaled by their standard deviation.

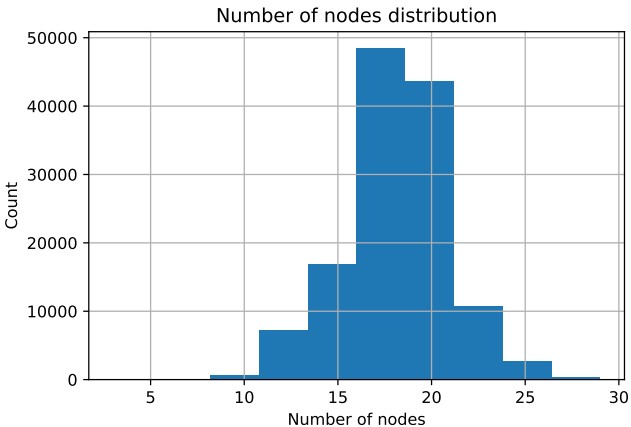

Figure 13: Distribution of the number of nodes in a graph for QM9 dataset.

**Benchmarks**
On the QM7 dataset, Quantum Machine benchmark reached MAE of 3.5 and 9.9 (Montavon et al., 2013).

The best models of the MoleculeNet benchmark (Wu et al., 2017) reached a test MAE of 2.86 ± 0.25 on QM7 and 2.4 ± 1.1 on QM9.

### D.3 DBLP_v1 AND NODE CLASSIFICATION

**Context**
DBLP_v1 is a graph stream built out of the DBLP dataset (Pan et al., 2013) consisting of bibliography data in computer science. To build a graph stream, a list of conferences from DBDM (database and data mining) and CVPR (computer vision and pattern recognition) fields are selected (as shown in Table 8). The papers published in these conferences are then used (in chronological order) to form a binary-class graph stream where the classification task is to predict whether a paper belongs to DBDM or CVPR field by using the references and the title of each paper.

Papers without references are filtered out. Then, the top 1000 most frequent words (excluding stop words) in titles are used as keywords to construct the graph (see Figure 14).

**Figures**
DBLP_v1 consists of 19456 graphs evenly split between the two groups of conferences (the two classes) from 2 to 39 nodes with an average of 10 nodes.

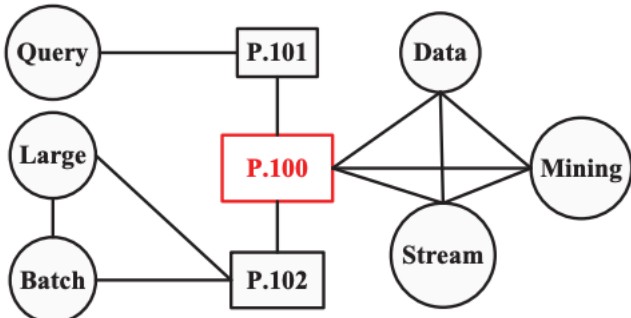

Figure 14: Graph representation for a paper (P.100) in the DBLP_v1 dataset. The rectangles are paper ID nodes and circles are keyword nodes from titles. The paper P.100 cites (connects) paper P.101 and P.102, and P.100 has keywords Data, Stream, and Mining in its title. Paper P.101 has keyword Query in its title, and P.102's title include keywords Large and Batch. For each paper, the keywords in the title are linked with each other

These graphs are actually local parts of a bigger graph. To perform node classification (on nodes representing a paper), the local neighborhood of each graph is extracted and a graph classification task is run.

There are 3 types of edges:

- 0: paper - paper
- 1: keyword - paper
- 2: keyword - keyword

Node features are only a unique ID to be identified between multiple graphs (cf P.100 in Figure 14 which will also appear in P.101 graph). There are 41325 unique IDs so keywords don't have a single keyword identifier among all graphs.

**Benchmarks**
DBLP_v1 (Pan et al., 2013) benchmarks show accuracy ranging between 0.55 and 0.80 in a chunked graph stream classification setup.

**Data augmentation**
The map from IDs to topics and paper IDs is also provided and has been used to perform data augmentation to provide node features. USing Stanford GloVe word embedding pre-trained on Wikipedia 2014 and Gigaword 5 in a 50-dimension space, each topic node could be enriched with its embedding. A boolean flag was also added to identify a node as a topic or not (a paper).

### D.4   COMPUTER VISION: LETTERS AND GRAPH CLASSIFICATION

**Context**
Letters datasets (Riesen & Bunke, 2008) are 3 datasets of distorted letter drawings with low, medium or high distorsion levels. Only the 15 capital letters of the Roman alphabet that consist of straight lines (A, E, F, H, I, K, L, M, N, T, V, W, X, Y, Z) are represented. Distorted letter drawings are converted into graphs by representing lines by undirected edges and ending points of lines by nodes.

**Figures**
Node attributes are their 2D positions and edges have no attribute. The graphs are uniformly distributed over the 15 letters. We focused on the medium distorsion dataset consisting of 2250 graphs.

**Benchmarks**
Benchmark results from k-NN are given by (Riesen & Bunke, 2008): 99.6% (low), 94.0% (medium), and 90.0% (high). The best classical algorithm we trained on this dataset was GraphSAGE with results of 100% (low), 94.5% (medium), and 80% (high)).

### D.5 GRAPHCOVERS AND THE WEISFEILER-LEHMAN ISOMORPHISM TEST

**Context**
Graph Neural Networks expressivity is related to the Weisfeiler-Lehman (WL) test. Recent work has been made to generate datasets of graphs undistinguishable by the WL test (Bamberger, 2022).

**Figures**
Using this work, we generated a small dataset of 6 non-isomorphic graphs of 21 nodes that can't be distinguished by MPNNs. These 6 graphs are then split in 3 arbitrary classes. The task at hand consists in being able to correctly distinguish the classes on the train data.

**Benchmarks**
Even on the train data, as they can't distinguish between the graphs, usual GNNs are not able to learn as shown in (Bamberger, 2022).

