# OpenReview forum: "Enhancing Graph Neural Networks with Quantum Computed Encodings"
_ICLR.cc/2024/Conference — ICLR 2024 Conference Withdrawn Submission_

### Official Review · Reviewer_8EzY · 2023-10-23

**Soundness:** 3 good
**Presentation:** 3 good
**Contribution:** 2 fair
**Rating:** 3
**Confidence:** 4

**Summary:**

The paper proposes novel ways to use quantum embeddings as positional encodings for graph transformers, and claims that these features can improve the performance and expressivity of existing graph neural network models. The quantum embeddings are generated by evolving the quantum system govern by the Hamiltonian which is parametrized by the graph structure. The paper also presents theoretical evidence to support that the quantum embeddings could distinguish some graphs that cannot be distinguished by the classical method (GD-WL test with RRWP embedding). The numerical experiments demonstrate that quantum embeddings can enhance state-of-the-art graph transformers on real-world and synthetic graph datasets.

**Strengths:**

1. The paper is well written and easy to follow. It introduces a framework for mapping graphs to quantum states and extracting quantum features such as correlations and probabilities from quantum dynamics.

2. Compared to the extensive prior work in quantum graph machine learning, the theory proposed in this paper is interesting. It shows that some quantum features are theoretically more expressive than classical ones, such as random walk probabilities, for certain classes of graphs.

3. It demonstrates that quantum features can enhance state-of-the-art graph transformers on standard benchmarks (though the gain is small), as well as on synthetic datasets that are hard for classical models.

**Weaknesses:**

1. From my point of view, in the literature of quantum graph learning, it is not particular novel to obtain the node/graph embeddings by encoding the graph structure into the Hamiltonian of the quantum system. Many previous papers have done a lot of research [1,2,3,4].
The authors need to provide a thorough and comprehensive summary of the existing relevant work in quantum graph learning.

2. The reviewer does not clearly see or understand where the main contributions of this paper lie in comparison to these highly relevant existing literature listed above. The reviewer believes that the research contributions of this paper is relatively insufficient, and the authors primarily transferred existing results to a new context, including some minor improvements, such as incorporating quantum embeddings into attention mechanisms.

3. The reviewer acknowledges that the numerical experiments reported in this paper are not particularly exciting. Most of the experimental results do not show significant improvements compared to state-of-the-art models especially for real-world benchmarks, or they only show marginal improvements.

4. It should be noted that there are some errors w.r.t the results in the tables. For the results for the CIFAR10 in Table 1, the accuracy reported by GRIT is 76.468 ± 0.881, which should be the top one. In Table 2, the std value of GRIT 2-QiQRW (0.023 ± 0.002) is relatively larger than the GRIT (0.023 ± 0.001), thus the best one should be the later.

5. Simulating the quantum evolution in classical computers is generally a hard problem. While these experiments may be potentially realized on real physical devices in the future, it would be beneficial if the authors could provide more detailed information on memory/CPU usage and training times. This would assist other researchers in assessing the feasibility of replicating the experiments.


[1] Tindall J, Searle A, Alhajri A, et al. Quantum physics in connected worlds[J]. Nature Communications, 2022, 13(1): 7445.

[2] Ye X, Yan G, Yan J. VQNE: Variational Quantum Network Embedding with Application to Network Alignment[C]//Proceedings of the 29th ACM SIGKDD Conference on Knowledge Discovery and Data Mining. 2023: 3105-3115.

[3] Henry L P, Thabet S, Dalyac C, et al. Quantum evolution kernel: Machine learning on graphs with programmable arrays of qubits[J]. Physical Review A, 2021, 104(3): 032416.

[4] Albrecht B, Dalyac C, Leclerc L, et al. Quantum feature maps for graph machine learning on a neutral atom quantum processor[J]. Physical Review A, 2023, 107(4): 042615.

**Questions:**

Please see the weaknesses.

---

### Official Review · Reviewer_17mU · 2023-10-31

**Soundness:** 2 fair
**Presentation:** 1 poor
**Contribution:** 2 fair
**Rating:** 3
**Confidence:** 4

**Summary:**

The submission introduces novel positional encodings tailored for graph transformers, inspired by quantum computing's long-range correlations. The quantum features, theoretically more expressive for specific graphs than traditional methods, are expected to enhance the performance of the Graph Inductive Bias Transformer (GRIT) using the Relative Random Walk Probabilities (RRWP) position embedding. Numerical simulations are conducted to exhibit the potential of the proposed encoding methods in boosting learning performance on multiple datasets.

**Strengths:**

In the realm of quantum computing, a pivotal challenge revolves around discerning which practical problems can genuinely benefit from the distinctive capabilities of quantum computers. This submission takes a step by exploring the potential of quantum computing to devise enhanced positional encoding methods, with potential implications for improving the performance of graph neural networks. The achieved results contribute to the ongoing exploration of quantum computing's applicability in practical problem-solving.

**Weaknesses:**

The primary weakness of the submission lies in its presentation. A substantial portion of the paper, five pages, is devoted to background information, covering quantum computing and quantum graph learning, while only 1.5 pages are allocated to introduce the proposed method. Consequently, the main text could benefit from a more balanced structure to ensure clarity in understanding both the rationale behind utilizing quantum computing for positional encoding and the practical implementation of the proposed encoding method. A revision of the main text is recommended to enhance the clarity of the contribution.

**Questions:**

1.	The motivation for exploring the ground states of local Hamiltonians, specifically the Ising and XY Hamiltonians, remains unclear in the submission. A clearer rationale for focusing on these Hamiltonians, such as their implementation efficiency on quantum computers or theoretical advantages, would provide context and enhance the understanding of the research direction.
2.	From a technical perspective, the proposed encoding methods can be seen as an extension of quantum approximate optimization algorithms (QAOAs). While this is a valid approach, it limits the submission's technical contribution, and a more distinct and innovative angle of contribution could be explored.
3.	The inclusion of numerical simulations to support Theorem 1 is unconventional for an academic paper. To improve clarity and maintain academic rigor, the authors might consider either removing the theorem or reducing the emphasis on theoretical achievements.
4.	The statement in Appendix B.1 regarding measuring multiple correlators to enrich the number of features extracted from the quantum state is somewhat confusing. As the operators $Z_iZ_j$ and $X_iX_j$ are commutable, the results $A_{ij}(\theta)$ obtained may not change significantly even with the removal of some features, which should be clarified for greater accuracy and coherence in the submission.

---

### Official Review · Reviewer_DnMk · 2023-11-01

**Soundness:** 2 fair
**Presentation:** 2 fair
**Contribution:** 2 fair
**Rating:** 3
**Confidence:** 4

**Summary:**

This paper proposes a framework to improve the performance of graph neural networks with QC-aided encodings. The authors provide an overview of existing research on graph transformers and quantum graph machine learning, and explain the theoretical aspects of their proposed framework. They introduce families of positional encodings based on quantum correlations and demonstrate their effectiveness through numerical experiments on standard benchmarks and large-scale datasets. The paper concludes by emphasizing the potential of leveraging quantum computing to enhance the performance of transformers for graph data.

**Strengths:**

This paper proposes a novel framework that uses quantum information to help the positional encoding. It explores the potentials of capturing the complex topological characteristics of the graph and encode them as quantum features that can be used as positional encodings in graph neural networks.

**Weaknesses:**

Scalability is not well discussed in the paper. It seems that the method requires a lot of quantum resources, which scales with the graph. Also, if the graph topology does not match the topology of quantum hardware, the implementation would require much more gates. The method in this paper may not be practical for large-scale applications in the near future. More information is needed for experiment settings on the quantum side.

**Questions:**

Why can't some quantum features be accessed classically?
In the framework of graph neural networks, the information is purely classical. I think the graph doesn't hold quantum-related nature, unlike tasks such as predicting the quantum states of a physical system.
From the data in Table 1, slight improvements can be seen. The model with quantum positional encoding does perform well, but it isn't remarkably better than classical methods. Is it expected to be so or?

---

### Official Review · Reviewer_1u4s · 2023-11-05

**Soundness:** 2 fair
**Presentation:** 1 poor
**Contribution:** 3 good
**Rating:** 5
**Confidence:** 3

**Summary:**

This paper presents a method for enhancing graph transformer models by introducing positional encodings derived from quantum long-range correlations, leveraging the computational strengths of quantum processing units. It theorizes and empirically shows that these quantum-inspired features can surpass the expressiveness of traditional encodings used in certain graphs. By providing empirical evidence of improved performance on benchmarks and large datasets, the work highlights the potential of quantum computing to advance the field of graph data analysis and transformer model performance.

**Strengths:**

The paper introduces a novel concept by applying quantum long-range correlations to enhance positional encodings in graph transformer models. This cross-disciplinary innovation has the potential to address limitations of current encodings and offers a theoretical framework for increased expressiveness, supported by empirical evidence on benchmarks and datasets.

While the approach is original, combining quantum computing with machine learning, the quality of the paper would benefit from further detailed methodological explanations to validate the robustness of the proposed model. Clarity is sufficient but could be improved, especially in articulating complex quantum concepts and their relevance to graph analysis.

The significance of the paper lies in its potential to impact graph data processing through quantum advancements, a noteworthy attempt but one that requires further exploration to fully ascertain its practical implications and to justify its purported advantages. Overall, the paper contributes an interesting perspective to the literature, yet it calls for a more cautious reception until the claimed benefits are consistently realized in practical applications.

**Weaknesses:**

The paper introduces a potentially transformative approach by integrating quantum mechanics into graph neural networks; however, it would benefit from addressing certain areas to enhance its contribution to the field. Firstly, the notations and variables such as \( \theta \), \( t \), \( \delta \), the adjacency matrix \( A \), and the feature matrix \( X \) require clear definitions within the specific context of the proposed model to improve the paper’s precision and replicability. This precision is particularly needed in Section 3.2.2, where the lack of detailed explanations may leave the reader uncertain about the implementation and significance of the variables used.

Moreover, the paper could strengthen its argument by elucidating the role of the quantum correlation matrix within classical graphical models. A clearer foundational explanation or intuition behind the effectiveness of this integration, especially in support of Theorem 1, would help articulate the purported advantages of the approach.

Finally, the comparative analysis presented in the results, while indicative of improvement, limits its benchmarking to the findings of Ma et al. (2023). Expanding this comparison to include a broader range of contemporary works could provide a more compelling argument for the proposed method's state-of-the-art status. Additionally, it would be constructive to contextualize the scale of improvement, explaining its relevance and potential impact, despite being modest.

Incorporating these enhancements would significantly solidify the paper's contributions and help establish the proposed method's place within the landscape of graph neural network research.

**Questions:**

Can you elaborate on how the quantum correlation matrix benefits classical graphical models and offer intuition behind the effectiveness of this approach?
What is the reasoning behind Theorem 1, and could you provide a more intuitive explanation or justification for its claims?